# Gating Mechanisms Underlying Sequence-to-Sequence Working Memory

## Abstract

Working memory is the process by which a system temporarily stores information across a necessary duration. Memory retention and manipulation of discrete sequences are fundamental building blocks for the underlying computation required to perform working memory tasks. Recurrent neural networks (RNNs) have proven themselves to be powerful tools for such problems, as they, through training, bring rise to the dynamical behavior necessary to enact these computations over many time-steps; a feat not obtainable using a finite memory system, such as transformer networks and reservoir computing. As of yet, the means by which these learned internal structures of the RNN result in a desired set of outputs remains broadly elusive. Furthermore, what *is* known is often difficult to extrapolate from due to a task specific formalism. In this work, we analyze an RNN, trained perfectly on a discrete sequence working memory task, in fine detail. We explain the learned mechanisms by which this network holds memory and extracts information from memory, and how gating is a natural architectural component to achieve these structures. A synthetic solution to a simplified variant of the working memory task is realized. We then explore how these results can be extrapolated to alternative tasks.

## 1 Introduction

Recurrent neural networks (RNNs) transform stimuli across multiple time-points to produce non-linear working memory representations that can be used to solve complex tasks (Elman, 1990; Hochreiter & Schmidhuber, 1997; Mante et al., 2013). Memorization and manipulation of discrete sequences of elements are a common low level requirement to many broad families of such problems (Hochreiter & Schmidhuber, 1997; Jordan et al., 2021; Yang et al., 2019). However, a full understanding of how the underlying dynamics learned by these networks accurately bring rise to the necessary computations remains an open area of research (Sussillo & Barak, 2013). That is, from a dynamical system's point of view (Guckenheimer & Holmes, 1983; Jordan et al., 2021), how does the network's internal phase-flow and attractor structures, brought forth by training, play a part in the found solution to the desired task? Furthermore, many of the known properties of trained RNNs' learned dynamical mechanisms are specific to individual problems (Henaff et al., 2016; Jarne, 2020; Ichikawa & Kaneko, 2021). Therefore, due to their narrow formalism, such attributes are often difficult to extrapolate to alternative tasks of the same family. It has been demonstrated

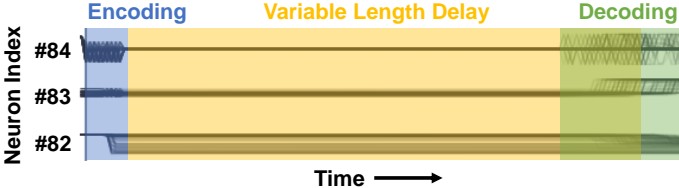

Figure 1: Superimposed trajectories for neurons $82-84$ across one thousand trials of VDCM, from a perfectly trained GRU. Neurons 82 and 83 demonstrate slow manifold dynamics during the delay period of each trial. These three neurons represent the qualitative behavior across all 250 neurons.

that the underlying attractor structures of RNNs successfully trained on an individual task are often similar, if not topologically equivalent, across networks, regardless of architecture, initialization, and hyper-parameters (Maheswaranathan et al., 2019). However, one would also expect commonalities between the underlying structures of RNNs trained on similar, but none the less, different tasks (Flesch et al., 2021; Yang et al., 2019) – a research feat not yet well explored.

To better train and interpret the solutions found by RNN models we require a finer grain analysis, where results can be broadened to very general classes of problems. (Karpathy et al., 2016) demonstrated a more in depth empirical exploration of GRU and LSTM architectures computing with sequential data, and identified different failure cases that can arise when training these models. However, this work studied the existence of underlying dynamical mechanisms indirectly from their effects on network output and single-neuron behavior, leaving out details on the functionality of each mechanism at the population level. If done, found mechanisms can be further studied and synthetically recreated to be made more understandable. Moreover, the synthetic realization of dynamical mechanisms, inspired by those obtained through gradient based optimization, can be combined and extrapolated from to form synthetic RNN solutions to related tasks.

We surgically analyze a single RNN trained on a discrete sequence working memory task. The network performs with no mistakes on a sufficiently sized validation set of trials. Inspired by the behavior discovered in the network, we design and experimentally validate a synthetic solution to a simplified version of the same task, realizable in relatively low dimensions. We then discuss how such findings apply to networks trained on different but mechanistically similar tasks, including a sequence-to-sequence translation task.

## 2 VARIABLE DELAY COPY MEMORY TASK AND RNN MODEL

In choosing an appropriate task, we look at *copy memory*; a standard benchmark to evaluate a neural network's ability to accurately recall information seen many time-steps in the past (Hochreiter & Schmidhuber, 1997; Henaff et al., 2016; Arjovsky et al., 2016). Let $A = \{a_i\}_{i=1}^K$ be a set of K symbols. We then pick $S, T \in \mathbb{N}$. The input is a vector of categories, length $T + 2S$, where each entry is one-hot encoded. Each trial of the task consists of three phases, of length $S, T$, and $S$ respectively. During the first phase (encoding), the network is presented with with S entries uniformly sampled from $\{a_i\}_{i=1}^K$, to be remembered sequentially. During the second phase (delay period), the network is fed $T - 1$ inputs of $a_{K+1}$, a *blank* category, indicating no important information is entering the network. At the final time-step of the second phase $S + T$, a delimiter $a_{K+2}$ is input to the network, indicating that the RNN should output the original $S$ entries of input in the same order by which they appeared in the first phase of the trial beginning at the next time-step (third phase – decoding). During these last $S$ time-steps, the inputs are all set to the blank category $a_{K+1}$. During the first $T + S$ time-steps (encoding and delay period), the network should output $a_{K+1}$. The task is to minimize the average cross-entropy of the outputs at every time-step. As such, the networks should remember a sequence of S elements for T time-steps.

Henaff, Szlam, and LeCun developed and experimentally verified a synthetic solution to this task (Henaff et al., 2016). However, if we allow $T$ to vary from trial to trial, the underlying dynamical mechanisms allowing the RNN to properly enact the computation remains elusive. We will refer to this task as *variable delay copy memory* (VDCM), as coined by (Henaff et al., 2016). We successfully trained a GRU network (Cho et al., 2014), by novel means (*Anonymous*), with a linear readout on VDCM, such that it performs perfectly on all the test trials. We were unable to train other network architectures on VDCM, including LSTM. For each trial we set $K = 8$, $S = 10$, and $T \sim \mathcal{U}(100, 101, ..., 120)$. The model used is represented as follows:

$$\boldsymbol{z}_t = \sigma(\mathbf{W}_z \boldsymbol{x}_t + \boldsymbol{b}_{iz} + \mathbf{U}_z \boldsymbol{h}_{t-1} + \boldsymbol{b}_{hz}) \tag{1}$$

$$\boldsymbol{r}_t = \sigma(\mathbf{W}_r \boldsymbol{x}_t + \boldsymbol{b}_{ir} + \mathbf{U}_r \boldsymbol{h}_{t-1} + \boldsymbol{b}_{hr}) \tag{2}$$

$$\boldsymbol{h}_t = (1 - \boldsymbol{z}_t) \odot \tanh(\mathbf{W}_h \boldsymbol{x}_t + \boldsymbol{b}_{ih} + \boldsymbol{r}_t(\mathbf{U}_h \boldsymbol{h}_{t-1} + \boldsymbol{b}_{hh})) + \boldsymbol{z}_t \odot \boldsymbol{h}_{t-1} \tag{3}$$

$$\boldsymbol{y}_t = \mathbf{V}_{out} \boldsymbol{h}_t + \boldsymbol{b}_{out} \tag{4}$$

$$y_{choice} = \arg\max([\boldsymbol{y}_t]_m), m \in [0, 1, ..., K] \subset \mathbb{N} \cup \{0\} \tag{5}$$

where $\boldsymbol{h}_t \in \mathbb{R}^d$, $d = 250$, $\boldsymbol{x}_t \in \mathbb{R}^{K+2}$, $\mathbf{W}_z, \mathbf{W}_r, \mathbf{W}_h \in \mathbb{R}^{d \times S}$ and $\mathbf{U}_z, \mathbf{U}_r, \mathbf{U}_h \in \mathbb{R}^{d \times d}$ are the parameter matrices, $\boldsymbol{b}_{iz}, \boldsymbol{b}_{ir}, \boldsymbol{b}_{ih}, \boldsymbol{b}_{hz}, \boldsymbol{b}_{hr}, \boldsymbol{b}_{hh} \in \mathbb{R}^d$ are bias vectors, $\odot$ represents element-wise

multiplication, and $\sigma(\boldsymbol{z}) = 1/(1 + e^{-\boldsymbol{z}})$ is the element-wise logistic sigmoid function. For the linear readout $\boldsymbol{y}_t \in \mathbb{R}^{K+1}$, $\boldsymbol{b}_{out} \in \mathbb{R}^{S-1}$ and $\mathbf{V}_{out} \in \mathbb{R}^{(S-1) \times d}$. $y_{choice}$ is taken as the largest element of $\boldsymbol{y}_t$ at each time-step, and represents the chosen class readout.

## 3 Encoding on Slow Manifolds

The computations required for VDCM fall into two main parts. The first is the memory structure. How does the GRU retain information about the elements presented to the network at each of the ten encoding time-steps? The second is memory recall. How does the GRU pull stored information from memory in the correct order? In this section we will focus on the former, and explain the structure by which our trained RNN uses to properly encode inputted information.

Fig. 1 demonstrates the behavior of hidden-state neurons $82 - 84$ (i.e. $[\boldsymbol{h}_t]_j$ for index $j \in [82, 83, 84]$) of the trained network, while performing VDCM. Let *neuron* refer to hidden-state neuron unless otherwise specified. The trajectory of these selected neurons across one thousand trials are superimposed, clearly indicating the three segments of the task. If we look at the delay period, beginning at $t = 11$, we notice that the neural activity appears to be near constant across each trial. The selection of neurons shown in Fig. 1 are representative of the behavior of across most neurons in the network, with the exception of several oscillatory modes that are rare and inconsistent across trials. A complete collection of neurons across trials can be found in the appendix E. What is of primary interest are the neurons that significantly vary trial to trail during the delay period, such as neurons 82 and 83. These varying neurons indicate the existence of a slow manifold, an observation in line with recent research (Ghazizadeh & Ching, 2021). A slow manifold is similar to an attractor (i.e. a fixed point, an attracting line, an attracting ring, etc.), where if the state of the system $\boldsymbol{h}_t$ lies on the attractor it will not change unless perturbed. However, the manifold is not entirely made up of fixed points, rather the speed of the phase flow in these regions is arbitrarily slow in a subset of directions. In the GRU architecture, such behavior results from either a pseudo-line attractor (Jordan et al., 2021), or from the influence of the update-gate $\boldsymbol{z}_t$ (Cho et al., 2014). If we look at the update-gate for dimensions with analogous behavior to 82 and 83, what we find is that, during the delay period, most demonstrate a value close to 1 for $\boldsymbol{z}_t$. Such activity simplifies equation 3 to the following approximate form:

$$[\boldsymbol{h}_t]_i \approx [\boldsymbol{h}_{t-1}]_i \tag{6}$$

where $i$ indexes the neurons with a high update-gate during the delay period. As such, the network retains near perfect memory of the past in these directions. We assume that the neurons' update-gates are the primary mechanism to enact slow-manifolds in our trained network. In the case of pseudo-line attractors, such slow flow is the result of the nullclines of the underlying continuous-time dynamical system, by which the network can be interpreted as a numerical approximation of, existing sufficiently close together in the hidden state-space (Jordan et al., 2021). However, the nullclines of this system cannot be oriented such that they form a pseudo-line attractor in any canonical direction (in the direction of a single neuron). We will see in the next section that the means by which our network is decoding are canonical in nature, and so we disregard this mechanism.

Given that this is the most likely mechanism used to encode information, how exactly this computation enacted? We trained this network to be able to encode $8^{10}$ possible sequences, $K = 8$ elements to choose from at each time-step, across $S = 10$ time-steps. If such a computation is implemented by carefully placing each trajectory on a slow manifold, the manifold can be segmented into regions where each individual readout element is outputted. Due to the implementation of the argmax function for VCDM, the largest element in the readout vector will be chosen as the class. A cartoon representation of such a regime is depicted in Fig. 2.

To empirically show that a slow manifold is the dynamical feature used and to determine how it is organized, we implement a perturbation based experiment. The low level details can be found in appendix A. We can assume an encoding time-step $q \in [1, ..., 10]$, and an element $p \in [0, ...7]$. If across two trials, we input a sequence where at time-step $q$ the element to be encoded is $p$, and an identical sequence where at time-step $q$ the element is not $p$, we can determine how the neuron representation differs after the encoding phase of each trial ($t = 11$). By comparing this difference over many sessions, we can determine which neurons are used to encode for this $q - p$ pair, how often each neuron is used, and approximate the expected value each neuron takes. We can then test the accuracy of our $q - p$ representation by inputting many sequences through the network, where

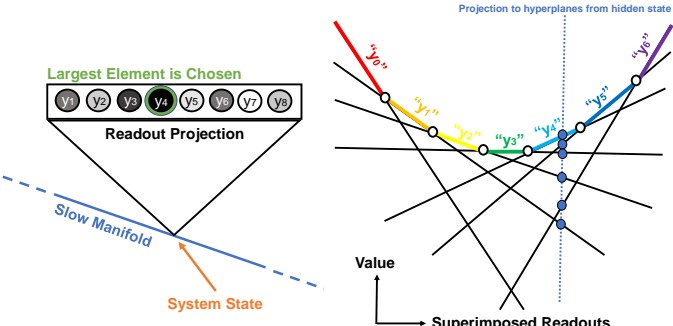

Figure 2: **Left:** the system state is on a slow manifold (blue). The output projection $y_t(h)$, can ensure that the a specified dimension in the output space has the largest value (green circle), Due to the *argmax* function, this dimension will be chosen as the class. **Right:** we can express each element of $y_t$ as a linear combination of the values of each neuron (defined by the learned output weights), which can be interpreted as a hyperplane, where our position on this hyperplane is determined by $h_t$. The hyperplanes can be arranged such that regions in the hidden-state space exists, where each possible readout takes on a higher value than the others. The white points indicate the class decision boundaries, and the blue points indicate each element of $y_t$ from an example $h_t$.

the elements at $t = q$ are not $p$, but we set, at $t = 11$, the $n$ most frequently used neurons in the representation to their approximate expected values for the representation. If the network output remains unaltered, but at decoding time-step $q$ the readout is incorrectly changed to $p$, we consider the trial a success. The results of this method are displayed in Fig. 3.

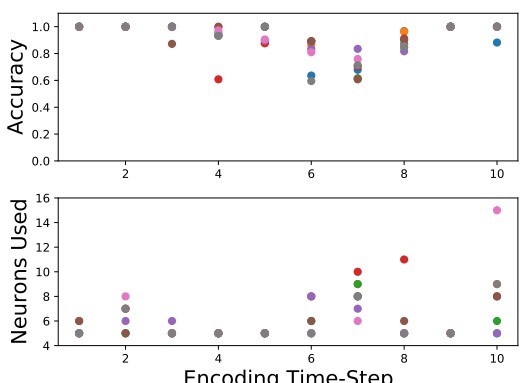

Figure 3: Results of our method to map out the structure of the slow-manifold in our trained RNN. **Top:** probability of a *successful* perturbation at the beginning of the delay period ($t = 11$) for each possible element, indicated by color, across each encoding time-step. *Success* indicates that the neurons perturbed successfully altered the appropriate decoding time-step readout to the desired element without altering the readouts at any other time-step. **Bottom:** number of neurons used for each perturbation. All pairs of time-steps and elements required at least 5 neurons.

Only 73 of the 250 neurons in the network *may* be used to encode memory. We can reorder these 73 neurons by their *center of mass* with respect to the 80 possible combinations of $q$ and $p$, where we sweep through all values $p$ can take on for $q = 1$, then $q = 2$, etc. Fig. 4 (Left) depicts such a reordering, revealing a block-like structure. We see that a nonempty set of neurons account for every element at each specific time-step $q$. The plot is colored by the expected value that neuron (row) takes when presented with a specific element at a given time-step (column).

To analyze the finer details of the manifold's structure, let's consider the neurons primarily tuned to a single time-step; those tuned to encode information at time-step 9, for example. We can use principal component analysis (PCA) to visualize the activity of these selected neurons in low dimensions (Bishop, 2006). We project down the state of the set of neurons tuned to time-step 9 at $t = 11$, across the set of test trials as demonstrated in Fig. 4 (Right). Data points are colored by which element $K$ was presented to the network at $t = 9$, and form eight separable clusters, one for each element. The points in each cluster vary in a single direction, indicated by the blue arrow. Since VDCM is a deterministic task, the only source for this variability is the influence of the input to the network at all time-steps preceding $t = 9$. This suggests that the neurons primarily tuned to encode information at $t = 9$ are not fully decoupled from the neurons tuned to other time-steps.

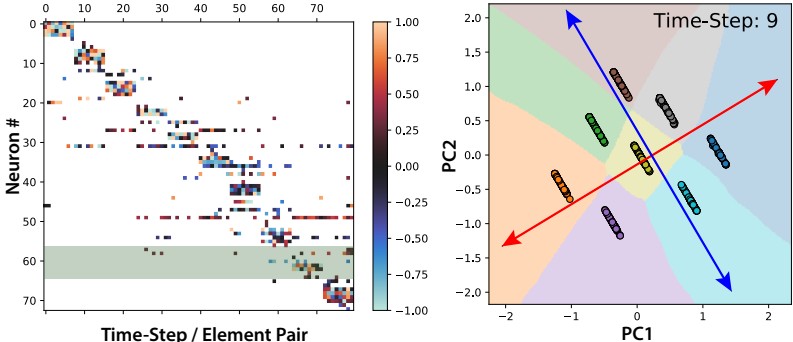

Figure 4: **Left:** the 73 neurons exhibiting slow manifold dynamics during the delay period. Every eight columns corresponds to the different elements that can be given at a specific encoding time-step. Matrix elements displayed in white indicate that the corresponding neuron (row) do not respond to changes in these element / time-step pairs. Further color indicates the expected value that neuron takes when a given element is input at the corresponding time-step. **Right:** PCA projection of the activity of the neurons tuned to encode information from $t = 9$ (**shaded region on the left plot**) at the beginning of the delay period ($t = 11$). The activity at $t = 11$ for each trial is plotted as a data point colored by which element was presented to the network at $t = 9$. The background coloring indicates what class is most likely to be readout at the ninth step of decoding, if that point in PC space was projected back up to the hidden subspace of neurons tuned to time-step 9 at $t = 11$, replacing them.

However, this observation brings about an important point regarding the neurons in our trained network that are primarily tuned to input presented at $t = 9$. While PCA brings about the dimensions with the highest variability across trials, it does not indicate which dimensions are most crucial for enacting the computation. While insightful in understanding how training brought forth various sub-mechanisms that make up the finer memory structures in the network, this direction allocated to previous inputs may not be used to indicate class during the ninth step of decoding. In the following section, we will dive into the second major computation required for VDCM, decoding. It will be shown that analysis of decoding will enable us to better understand the mechanism used for encoding. For all other encoding time-steps, plots analogous to Fig. 4 (right) can be found in appendix B.

## 4 DECODING AND THE ROLE OF GATING

How does information from memory get readout in the desired order? The readout at each time-step is determined by equation 4 and equation 5. The linear readout matrix $\mathbf{V}_{out}$ and readout bias vector $\boldsymbol{b}_{out}$ separate the hidden-state space $\boldsymbol{h}_t$ into classification regions with linear decision boundaries. Prior to receiving the readout signal, the network remains in a regime where only the delimiter symbol "8" is outputted. The readout signal perturbs the hidden-state of the network into a *decoding subspace*, where some mechanism for determining which decoding time-step the network is on must activate (i.e a *clock*). Decoding the first element of the sequence is easy enough, as the network needs only to perturb the hidden-state to a region where the neurons tuned to encoding the first time-step of each trial are dominant in readout (Fig. 2). But then how does the network properly output the remaining elements of the input sequence? The most natural means to enact this behavior is built into the GRU architecture, as the added gates, $\boldsymbol{z}_t$ and $\boldsymbol{r}_t$, act to *forget* or *reset* information stored in the hidden-state. If some neuron, of index $\rho$, in the network has corresponding update-gate and reset-gate values close to zero, the evolution of that neuron can be approximated as follows:

$$[\boldsymbol{h}_t]_\rho \approx \tanh([\mathbf{W}_h]_\rho \cdot \boldsymbol{x}_t + [\boldsymbol{b}_{ih}]_\rho) \tag{7}$$

where $[\mathbf{W}_h]_\rho$ indicates row $\rho$ of $\mathbf{W}_h$. Therefore, the network loses all information previously stored in this neuron of the system (direction of the hidden state-space). Moreover, the neuron is overwritten by the input $\boldsymbol{x}_t$.

We want to analyze our trained GRU network to see if the decoding regime exploits this behavior. Prior to doing this, it is important to define what exactly is meant for a neuron to be "reset." We

will define the metric $\kappa = \overline{(1 - [z_t]_\rho)(1 - [r_t]_\rho)} \in [0, 1]$, where the overhead bar represents a sample mean. At a given time-step, if a neuron $\rho$ is to forget all previous information held within it $(1 - [z_t]_\rho)(1 - [r_t]_\rho) \approx 1$. The closer to zero this term is, the more prior information is retained. We run 1000 randomly generated input sequences through the network and determine $\kappa$ for every neuron at each decoding time-step. We then determine a threshold $\zeta$, where neurons with $\kappa > \zeta$ at a given time-step are "reset." We chose $\zeta = 0.255$, and detail our reasoning in appendix C.

Using this threshold, we reorder the neurons of the network by which decoding time-step each is reset after for the first time (i.e. the first time that the expression $\kappa > \zeta$ is true after the readout signal is presented). Fig. 5 (Left) depicts this reordering, allowing us to visualize $\kappa$ for each neuron using our choice of $\zeta$. We notice a clear staircase pattern, indicating the existence of a non-empty set of neurons that reset for the first time after every time-step of decoding. Furthermore, due to the methods used to obtain our value of $\zeta$, it is assured that at least one neuron in each of these sets is tuned to the equivalent time-step during the encoding phase of each trial.

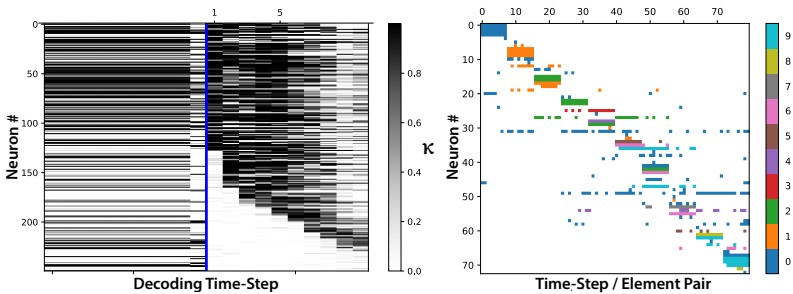

Figure 5: **Left:** mean activity of our defined metric $\kappa = \overline{(1 - [z_t]_\rho)(1 - [r_t]_\rho)}$ for each neuron when we line up the decoding phases across trials. We reordered the neurons such that those which "reset" at time-step 1 of decoding are on top, followed by those which "reset" for the first time at time-step 2, etc. Doing so reveals the demonstrated staircase pattern, indicating that there exists a distinct set of neurons which are reset at each step of the decoding phase. **Right:** a recoloring of Fig. 4, indicating at what time-step each neuron active in memory retention during the delay period is reset in the decoding phase. We note that every time-step of encoding has at least one corresponding neuron that is appropriately reset at the equivalent step of decoding, with the exception of those tuned to $t = 3$.

We recolor Fig. 4 (left) by which time-step of decoding each of the 73 neurons involved in encoding are reset after. This is demonstrated in Fig. 5 (Right). For the set of neurons tuned to each time-step of the encoding phase, at least one neuron is reset after its corresponding decoding time-step. This isn't a surprise, as we ensured this was the case when choosing $\zeta$. However, what is surprising is that for each set of neurons, with the exception of $t = 3^*$, at least one neuron exists that is tuned to every element of its corresponding time-step. Therefore, this regime demonstrates a near complete picture of the most important neurons responsible for performing VDCM. We say *important* to indicate that these neurons are the primary carriers of information (i.e. hold memory) throughout each trial of the task, while the others play a more supporting role; either managing internal mechanisms for telling time during the encoding/decoding phases of the task or fine tuning the discussed dynamical mechanisms, such as the direction allocated for noise demonstrated in Fig. 4 (right). Furthermore, our analysis suggests that the system is primarily canonical in its encoding and decoding mechanisms. This is a result of the GRU architecture, as both gates are canonical in their functionality, meaning information is maintained or reset in the directions of individual neurons (Cho et al., 2014). Furthermore, the internal clock is finely tuned to ten time-steps. We have tested if the network can generalize to different values of S. If we vary S, not a single trial's readout is entirely correct. If $S < 10$, the first S elements are properly encoded/decoded. If $S > 10$, the first 10 elements are properly encoded/decoded.

---

*Information from $t = 3$ is reset after the fourth time-step of decoding, for some values of K, suggesting possible intersection between the regions of the slow manifold, and or that the geometry of theses regions are more complex, requiring the neurons tuned to additional time-steps to aid in enacting the computation.

## 5 SOLUTION TO THE $K = 2$ CASE OF VDCM

Given what we know about the inner functionality of our trained network, we'd like to develop a complete solution for performing VDCM based off of found behavior. The solution should layout a specific set of parameters for the GRU network. No training will be required and the internal dynamical behavior must be interpretable. We will incorporate a slow manifold to encode the input sequence. Furthermore, we will design an appropriate readout projection and sequentially reset the neurons allocated to memory as to output each element of the sequence in the desired order. This will be done with a separately designed clock mechanism, which will be used for determining the number of time-steps which have passed during both the encoding and decoding phases of each trial.

We restrict our solution to the $K = 2$ case of VDCM, such that each sequence to be encoded is a binary string of length $S$. Any information expressible as a binary string of elements can be encoded and later decoded in this context, with $2^S$ different states achievable with a single network of $2(S+1)$ neurons, as we shall demonstrate below. Furthermore, this allows for a more pedagogical approach in our presentation, as our goal is to build a deep intuition for these mechanisms and demonstrate a proof of concept in extrapolation. As we'll see, only one neuron is needed to hold memory from a given time-step in the $K = 2$ case of this problem. Fig. 6 (left) depicts a schematic representation of our solution. For easy visualization, Fig. 6 (right) illustrates the decoding mechanism for the $S = K = 2$ case.

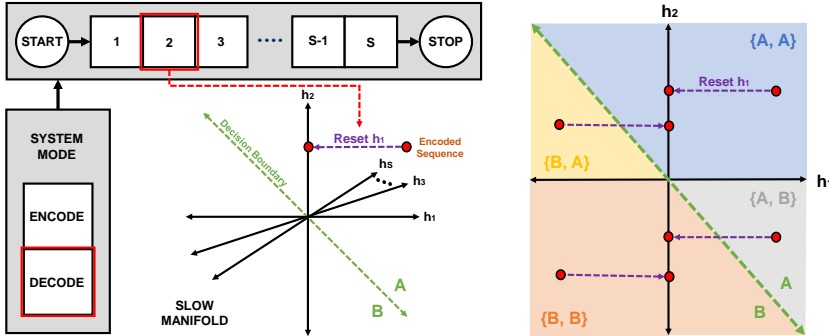

Figure 6: **Left:** Schematic representation of the synthetic solution to VDCM for arbitrary $S$ and $K = 2$, where elements are notated as either $A$ or $B$. The system acts in one of two modes, either encoding or decoding. During the encoding phase of each trial, elements to be remembered are presented to the network, setting the values of the first $S$ neurons (a slow manifold) sequentially, as dictated by an internal clock. After the first $S$ time-steps, the system switches out of the encoding mode into decoding. The clock is not reactivated until the readout signal is given, which will begin resetting the values of the $S$ neurons allocated to memory to zero, again sequentially. Done properly, the network will readout the encoded elements in their original order. **Right:** An example of the memory structure for the $S = K = 2$ case of our solution. The decision boundary separates the $h_1 - h_2$ plane into regions associated with each element. Four subregions exist, where information can be encoded such that each of the four possible sequences can be readout.

We begin by reiterating what each dimension of our input $\boldsymbol{x}_t \in \mathbb{R}^4$ describes. $x_1$ and $x_2$ are the two possible sequence elements, $A$ and $B$ respectively, $x_3$ is the blank symbol and $x_4$ is the readout signal. All unspecified parameter matrices and bias vectors only contain zeros in their entries. Fig. 7 depicts a complete picture of all nonzero parameter matrices and bias vectors used in our synthetic solution. From this, we can further define each subcomponent. We allocate $S$ neurons initialized at zero to memory, and $S + 1$ neurons, all initialized at $-1$ except for the second to last, which is initialized at 1, to the clock. Furthermore we allocate one neuron initialized at $-1$ to a switch. The switch will determine if the system is finished encoding or not, by saturating near $-1$ (low) or 1 (high). We also choose two parameters, $\alpha$ and $\beta$, such that $\alpha \gg \beta \gg 0$. $\alpha = 10$ and $\beta = 3$ has worked well for us. All unspecified subcomponent entries are considered to be zero.

Starting from the upper left, $\lambda_{11}$ encodes information directly into memory. Each row $i$ of $\lambda_{11}$ is $\begin{bmatrix} r^{-i} & -(r^{-i}) \end{bmatrix}$. We chose $r = 3$. The larger $r$ is, the greater in value $S$ can be while maintaining perfect accuracy on the task. $\lambda_{21}$ ensures the clock doesn't indicate two simultaneous time-steps

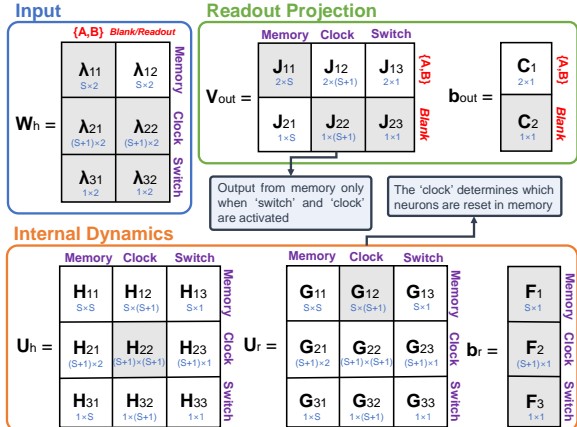

Figure 7: Nonzero parameter matrices and bias vectors used in our synthetic solution to VDCM. Each is broken up into subcomponents, where nonzero subcomponents are denoted in gray. The dimensions of each subcomponent are depicted in blue. The functional connectivity of each block matrix can be obtained by the purple and red labeling. For example, consider $\mathbf{V}_{out}$, which takes part in the readout projection from the hidden-state. Whether an $A$ or a $B$ is outputted is dependent only on the hidden-state dimensions allocated to memory.

during the encoding phase. Every element in the last two rows of $\lambda_{21}$ are $-\alpha$. The submatrix $\lambda_{31} = [-\alpha \quad -\alpha]$ holds the switch low during the encoding phase. The last row of $\lambda_{22}$ is $[-\alpha \quad \alpha]$, ensuring the clock remains deactivated during the delay period and reactivates upon receiving the readout signal. $\lambda_{32} = [\alpha \quad \alpha]$ keeps the switch high after the encoding phase.

In regards to the hidden-state dynamics, $H_{22}$ has $\alpha$ for every entry along the super-diagonal. This is the clock mechanism, which works in isolation, causing the clock neurons to sequentially rise from a low to a high value. Every entry of the bias sub-blocks $F_2$ and $F_3$ are set to $-\beta$. $F_2$ ensures that after a neuron allocated to the clock takes on a high value, it will be brought back to a low value at the next time-step. We chose $\beta \ll \alpha$ to ensure that neurons multiplied by $\alpha$ can overcome the functional effects of those multiplied by $\beta$. Similarly, $F_3$ keeps the neuron allocated to the switch low during the encoding phase. The entries $\sum_{j=0}^{S-1}(S-j)$ for each $j$ column of $G_{12}$ are all $\alpha$, and act to sequentially reset each of the $S$ neurons allocated to memory after each time-step dictated by the clock during the decoding phase. Every entry of $F_1$ is $\beta$, as to ensure the reset-gate for each neuron in memory is kept high prior to decoding, thereby retaining memory.

The primary component to readout from memory is $J_{11}$. The first row of $J_{11}$ is $[1 \quad 0 \quad 1 \quad 0\ldots]$, always alternating. The second row takes a similar structure and is $[0 \quad -1 \quad 0 \quad -1\ldots]$, again alternating. In conjunction with the placement of the encoded information on the slow manifold, as defined by the two geometric series which make up the columns of $\lambda_{11}$, this ensures that information encoded at earlier time-steps takes precedence until reset. Fig. 6 (right) depicts the resultant decision boundary when $S = 2$. All but the far right entry of $J_{22}$ is $-\alpha$, $J_{23} = -\alpha$, and $C_2 = \frac{3-2S}{2}\alpha$. These ensure that, unless both the switch is high and the clock is activated, only the blank symbol will be outputted. Thus completes our synthetic solution to the $K = 2$ case for VDCM. We validated this solution on the same test set of trials used for our traditionally trained network, and it performs with perfect accuracy. Examples of the hidden-state dynamics for the $S = 2$ and $S = 10$ cases are depicted in Fig. 8, and the code to generate these plots are in the supplementary material.

## 6 DISCUSSION

We've identified and analyzed a set of underlying dynamical mechanisms which enact the computations necessary to perfectly perform VDCM, a sequence-to-sequence working memory task, with a GRU network architecture. The internal dynamical behavior of the RNN constitutes the execution of both an encoding mechanism, where information can then be held indefinitely in memory, and a decoding mechanism, which makes use of and deletes information held in memory for accurate readout. The encoding regime stores information on a slow manifold, which makes up a subspace of the hidden-state space of the system. Regions of this manifold are tuned to specific time-steps of the encoding period, where subregions within each region are structured to differentiate between what input was presented to the network at the time-step the region was tuned to. The decoding mechanism deletes information sequentially from memory, allowing data stored at later time-steps to take precedence in the readout. Both of these mechanisms take advantage of gating to behave properly. We then detailed out a synthetic solution to the $K = 2$ case of VDCM and demonstrated its func-

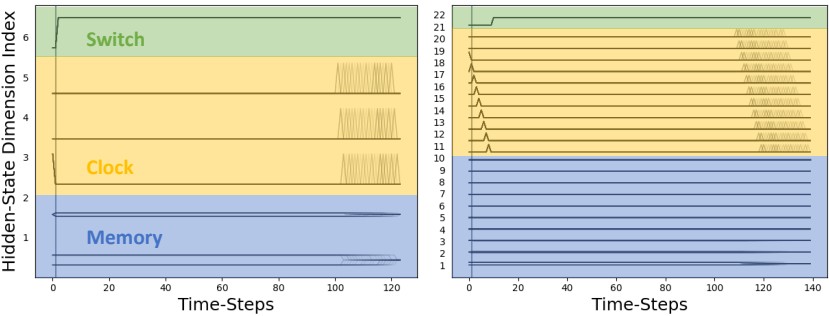

Figure 8: Hidden-state trajectories for 1000 superimposed trials of VDCM using our synthetic solution, for both the $S = 2$ case (left) and the $S = 10$ case (right). Neurons fall into one of three functional categories. $S$ neurons are allocated to holding memory. $S + 1$ neurons create the clock, which indicates what time-step of encoding or decoding is present, and one neuron is allocated to a switch. When *low*, the switch indicates the trial is in its encoding phase. When *high*, the trial has finished encoding. A high switch in conjunction with the readout signal begins the decoding phase.

tionality. In this section, we want to convince the reader that this analysis does not so narrowly apply only to RNNs trained on VDCM, but on alternative tasks as well.

The mechanisms realized in our synthetic solution can be applied in a variety of different contexts, so long as the mechanistic function required of each is retained across tasks. Consider the parity bit problem (Wegener & Pruim, 2005). The RNN is given a binary string of length $S$ and has to simultaneously output a binary string, indicating if the number of 1's presented to the network is *even* or *odd*. Unaltered, this task only requires an XOR operation between the input and output at the previous time-step. This suggests the need of a memory structure (or buffer) that holds one piece of information – the last element outputted. However, if we change the task so that the RNN needs to readout the desired binary sequence arbitrarily long after then entire input sequence has been presented, then the task is nearly identical to VDCM (Fig. 9 (middle)). The one caveat is that the information encoded in memory is not the input itself, but the output from the XOR operation of the input and information about the last element stored in memory. Let's construct another task; a

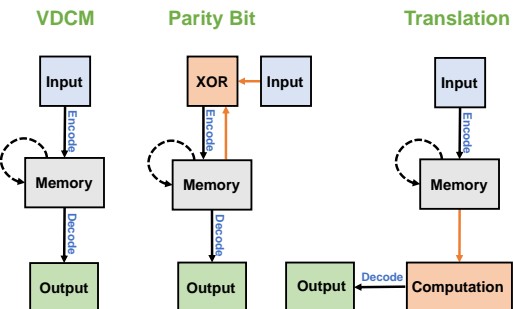

Figure 9: Schematic representation of the mechanistic connectivity of the processes and structures used to accurately perform VDCM, modified parity bit, and a binary character translation task. **Left:** input is encoded in memory, which is self sustaining, and is later decoded and outputted. **Middle:** nearly identical to VDCM, but what is encoded is the output of an XOR operation between the input and the previous encoded value in memory. **Right:** same as VDCM, but memory must undergo a nonlinear computation prior to decoding.

sequence-to-sequence translation task with a binary character language and a variable delay between receiving the input binary sequence and needing to output a desired binary sequence. Suppose that if the input sequence ends in a 1, the desired output sequence is the input sequence in reverse order, beginning with the last element seen. Furthermore, if the sequence ends in a 0, the desired output sequence is the input sequence with all the bits flipped – every 0 becomes a 1 and vice versa. Such a task is also nearly identical to VDCM. The memory structure can be made the same, but must undergo some nonlinear computation prior or during each decoding time-step (Fig. 9 (right)). We were unable to train an RNN on this latter task by any conventional means (appendix D), suggesting a need for more sophisticated training methods which leverage the dynamical behavior required of the RNN to complete the task. We leave this to future work, and note this result as an important example of why the work done in this manuscript is important for RNNs when performing more sophisticated tasks.

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

# A    SLOW MANIFOLD MAPPING

---

Algorithm 1: Slow Manifold Map

---

1: **procedure** MAP$(S, K)$
2:     $Q \leftarrow \{0, 1, ..., S - 1\}$                                                                ▷ time-steps
3:     $P \leftarrow \{0, 1, ..., K - 1\}$                                                              ▷ available symbols
4:     **for** each element $i$ of Q **do**
5:         **for** each element $j$ of P **do**
6:             $q \leftarrow Q_i$
7:             $p \leftarrow P_i$
8:             **for** $N$ iterations **do**                                                    ▷ we chose $N \leftarrow 1000$
9:                 Generate $\gamma$                            ▷ random input sequence to the RNN where $\gamma_q \neq p$
10:                 $\nu \leftarrow \gamma$
11:                 $\nu_q \leftarrow p$
12:                 Run both $\gamma$ and $\nu$ through the RNN and save the hidden-states of each at $t = S$
13:                 Record which $k$ neurons are sufficiently different  ▷ threshold of $10^{-3}$ was used
14:             Approximate $\mathbb{E}[\boldsymbol{h}_k]$ when $\nu$ is inputted to the RNN $\forall k$
15:             **for** M iteration **do**                                  ▷ we chose $M \leftarrow 1000$ and $n \leftarrow 4$
16:                 Generate $\tilde{\gamma}$                         ▷ random input sequence to the RNN where $\tilde{\gamma}_q \neq p$
17:                 Run $\tilde{\gamma}$ through the RNN, but alter $\boldsymbol{h}_k = \mathbb{E}[\boldsymbol{h}_k]$ for the $n$ most frequently recorded
                    neurons at $t = S$ during the forward pass
18:                 **if** $y_{choice} = p$ at step $q$ of decoding and all other outputs are correct **then**
19:                     record that this iteration was a *success*
20:             **if** number of successes is less than $M$ and $n \leq n_{max}$ **then**   ▷ we chose $n_{max} \leftarrow 15$
21:                 $n \leftarrow n + 1$
22:                 **go to** 15
23:             **else if** $n > n_{max}$ **then**
24:                 Disregard all but the $n$ recorded neurons for $n$ which gave the most successes
25:             **else**
26:                 Disregard all but the $n$ recorded neurons
        **return** The remaining values for each set of recorded neurons for all $qp$-pairs

---

# B ALL PCA PLOTS

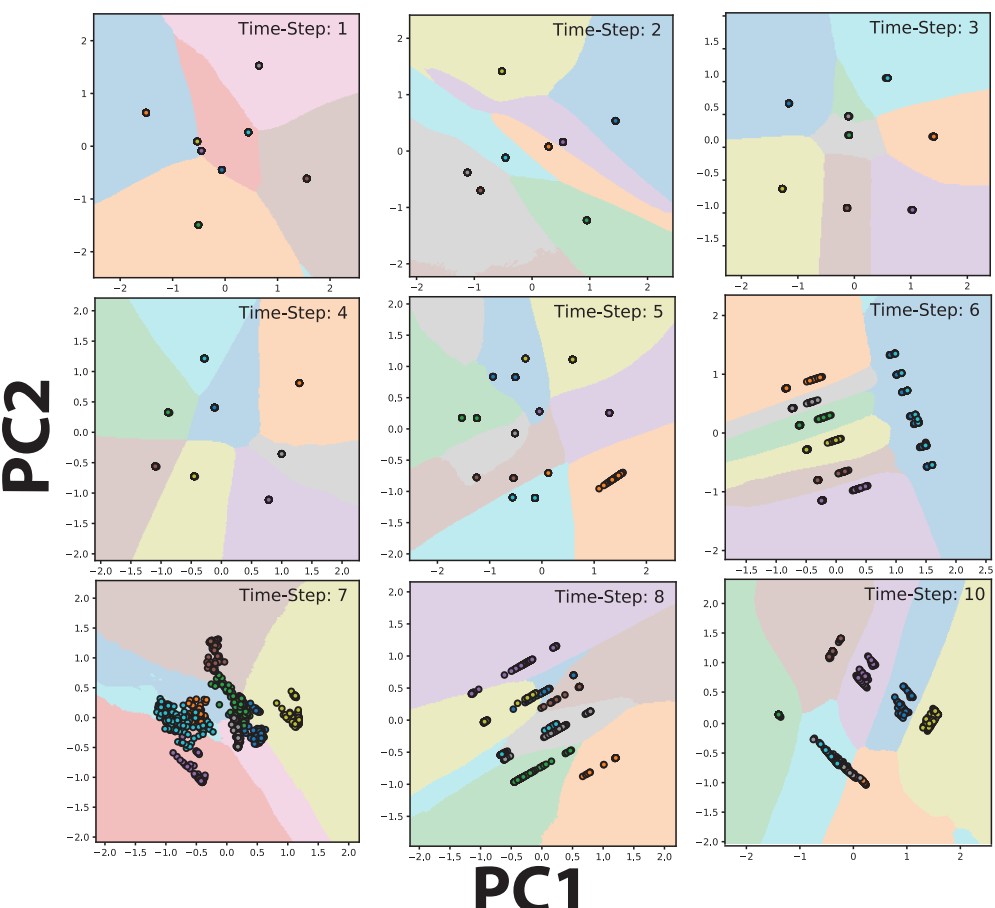

Figure 10: Equivalent plots to Fig. 4 (Right), for each set of neurons used to encode each additional time-step during the encoding phase of VDCM.

# C RESET THRESHOLD

We'd like our threshold $\zeta$ to be relatively low (i.e close to zero), where $\kappa$ exhibits little variance about $\zeta$. Furthermore, we need to ensure that a nonempty set of neurons exists for each set of neurons tuned to each encoding time-step, such that each is reset for the first time during the correct decoding time-step. For example, if a neuron is tuned to encode information presented at $t = 5$, then when the decoding period of the trial begins, the neuron should remain "not-reset" up until the fifth time-step of readout, where it will then properly reset. This will also indicate which neurons described in section 3 are most important for holding onto memory at each encoding time-step. Fig. 11 further details our reasoning for a choice of $\zeta$, which we set to $0.255$.

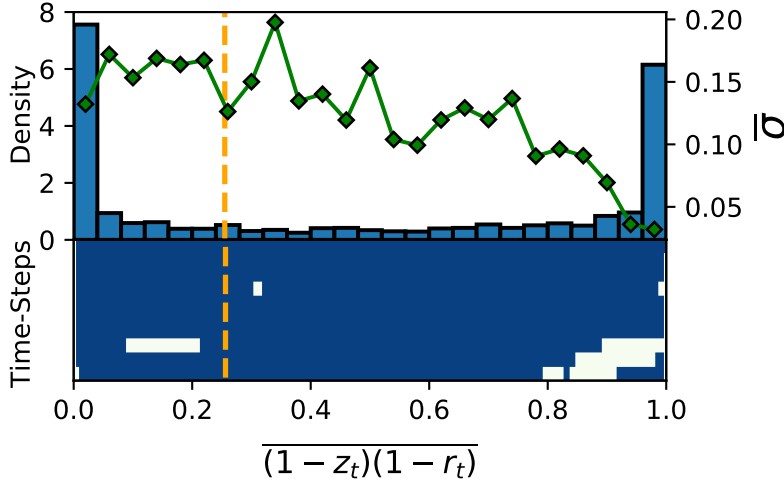

Figure 11: Plot indicating how we chose a threshold $\zeta$ on $\kappa$, where values under this threshold are considered "not-reset", and those above are considered "reset." **Top:** histogram showing the sample mean values of $\kappa$ each neuron takes across all steps of the decoding phase, over one thousand trials. Notice the bimodal distribution, as this metric for most neurons is usually close to zero or one. The green curve represents the standard deviation of the average values of $\kappa$ for each bin of the histogram. We want to pick a $\zeta$ with relatively low variance to ensure that individual neurons do not cross back and forth over the threshold trial to trial. **Bottom:** For varying $\kappa$, white indicates values of $\zeta$ where no neurons tuned to a given time-step in the encoding phase are shown to reset by our analysis. We want our threshold to not only be small, but to contain at least one neuron for each encoding time-step to be reset appropriately.

## D  TRAINING RESULTS

Hyper-parameters were chosen using Bayesian optimization (Ax https://ax.dev/ from the Ray Tune library) (Liaw et al., 2018) with validation likelihood used as the decision criterion. Specific values for the learning rate, learning rate scheduler, learning rate decay factor and gradient clipping were chosen over 100 iterations of the tuning algorithm All models were trained using Adam, with a batch size of 200; models used for hyper parameter optimization were trained for at most 100 epochs, while the models that were ultimately chosen were trained for a fixed 200 epochs. Standard cross-entropy loss was used as the loss functional. Unless implied otherwise by the name, models were initialized using the default method in PyTorch, which internally samples from a uniform distribution $\mathcal{U}(-\frac{1}{\sqrt{N}}, \frac{1}{\sqrt{N}})$ with $N$ being the number of hidden units.

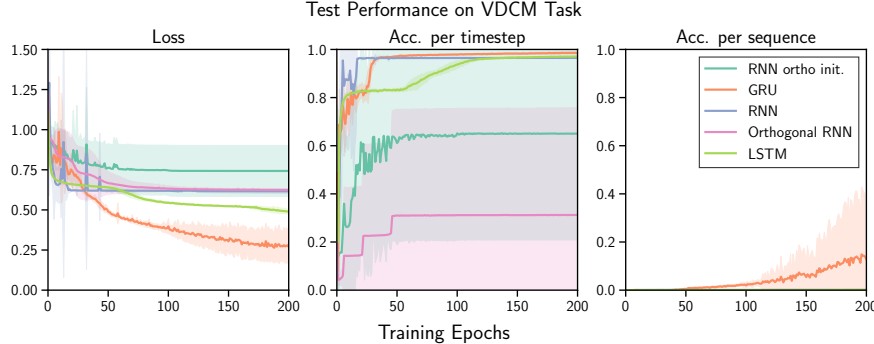

Figure 12: Test performance for trained RNNs on VDCM. Colors indicate mean activity of each architecture and initialization strategy used (10 networks each). The shaded region about each curve depicts the variance across networks. **Left:** cross-entropy loss across epochs. **Middle:** traditional network accuracy. **Right:** depicts what fraction of test trials the network outputted the sequence with no mistakes. Notice that no network other than GRU got close to learning VDCM.

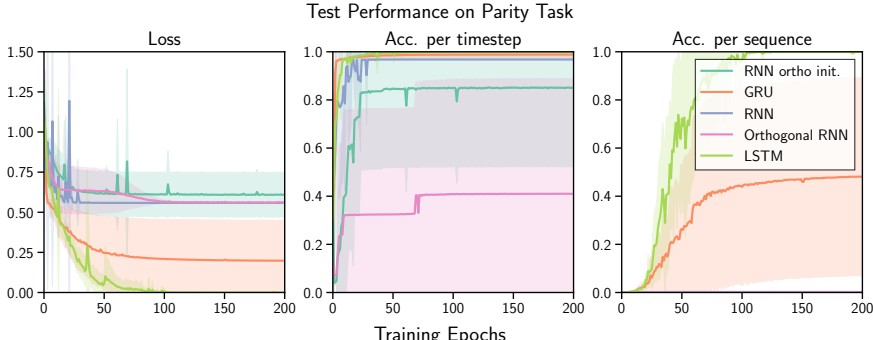

Figure 13: Test performance for trained RNNs on a modified parity bit task. Colors indicate mean activity of each architecture and initialization strategy used (10 networks each). The shaded region about each curve depicts the variance across networks. **Left:** cross-entropy loss across epochs. **Middle:** traditional network accuracy. **Right:** depicts what fraction of test trials the network outputted the sequence with no mistakes. LSTM does manage to learn this task.

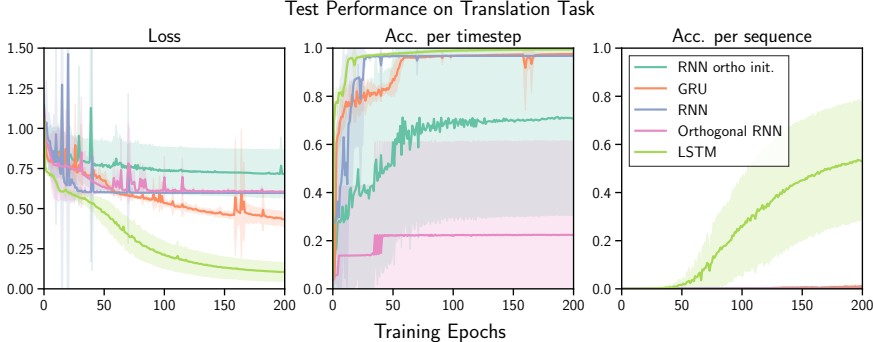

Figure 14: Test performance for trained RNNs on a binary character translation task. Colors indicate mean activity of each architecture and initialization strategy used (10 networks each). The shaded region about each curve depicts the variance across networks. **Left:** cross-entropy loss across epochs. **Middle:** traditional network accuracy. **Right:** depicts what fraction of test trials the network outputted the sequence with no mistakes. LSTM learns this task the best, but is still unable to consistently output entire the desired sequences with no mistakes.

# E TRAINED NETWORK HIDDEN-STATES ACROSS TRIALS OF THE TEST SET

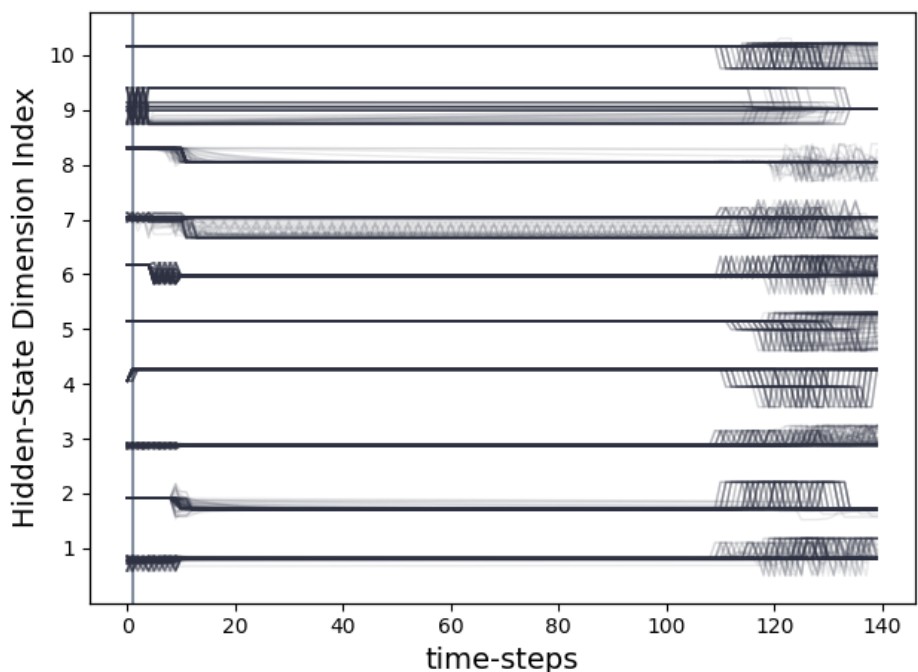

Figure 15: Superimposed trajectories one thousand trials of VDCM, from a perfectly trained GRU.

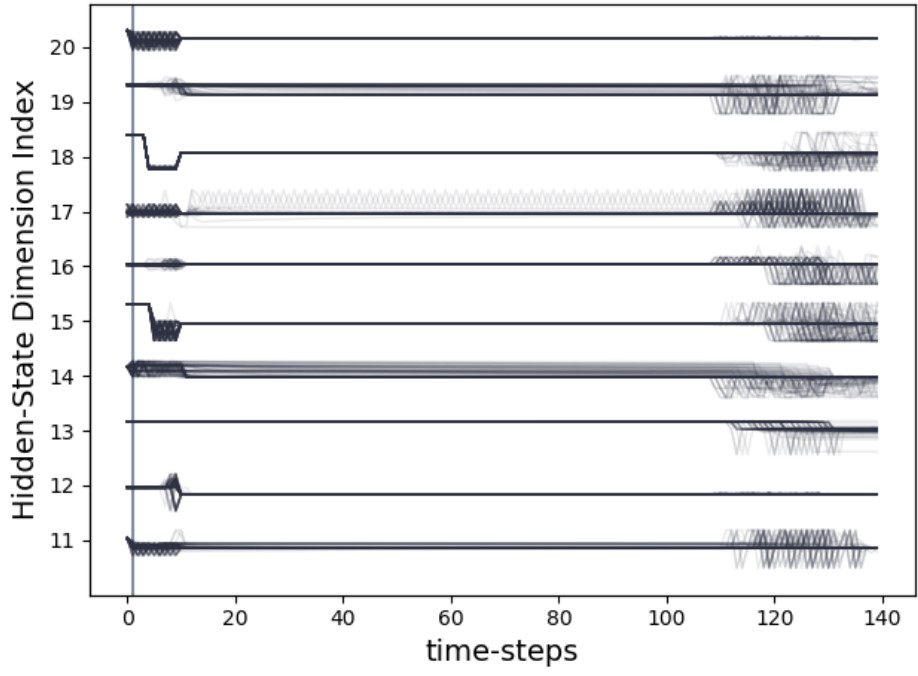

Figure 16: Superimposed trajectories one thousand trials of VDCM, from a perfectly trained GRU.

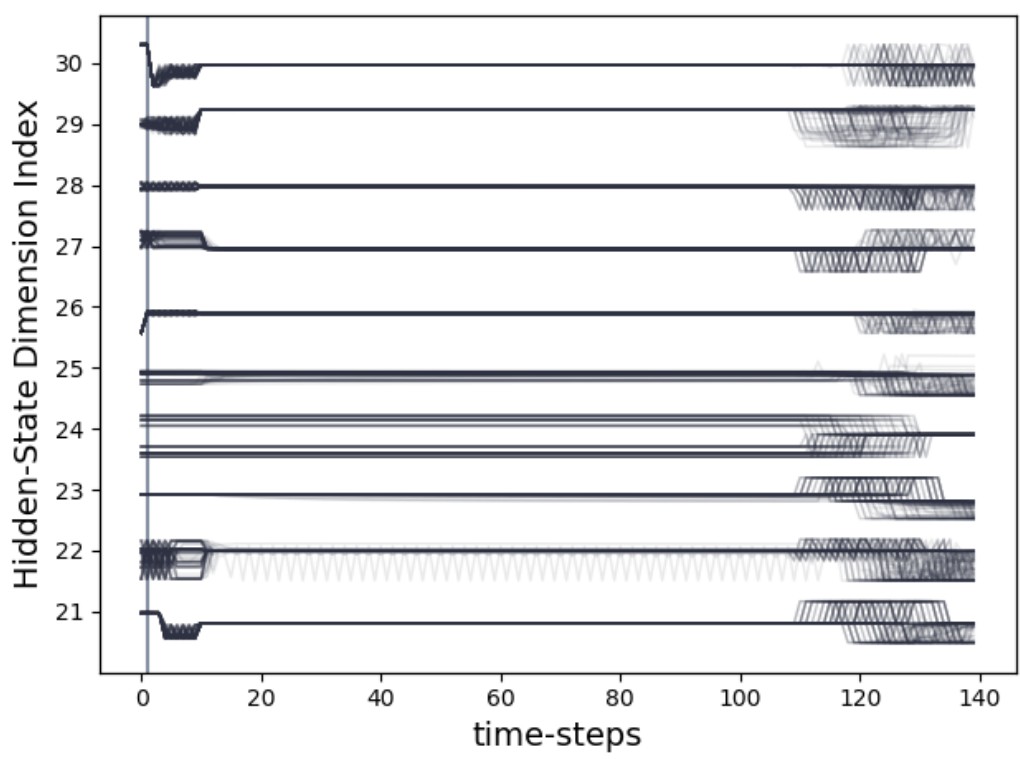

Figure 17: Superimposed trajectories one thousand trials of VDCM, from a perfectly trained GRU.

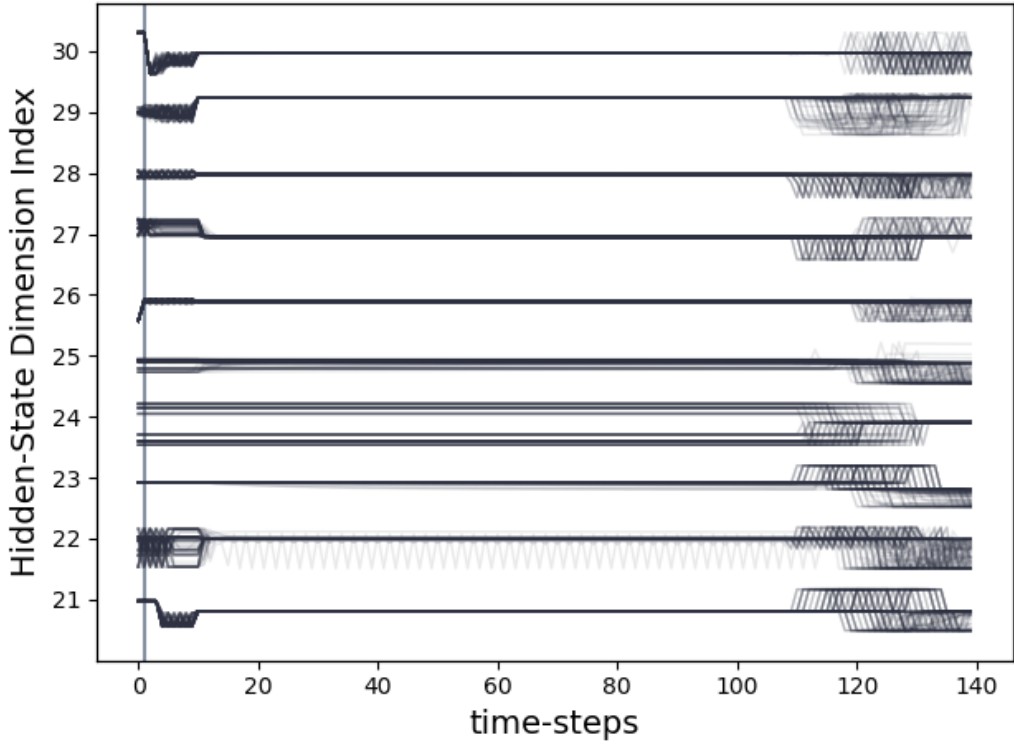

Figure 18: Superimposed trajectories one thousand trials of VDCM, from a perfectly trained GRU.

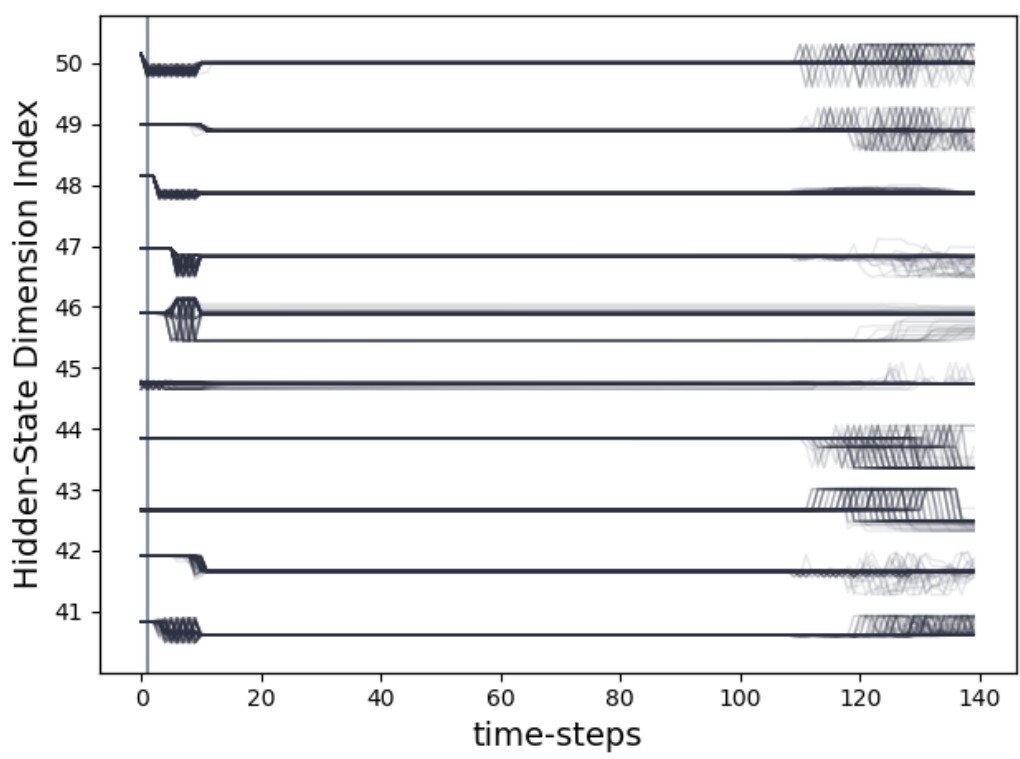

Figure 19: Superimposed trajectories one thousand trials of VDCM, from a perfectly trained GRU.

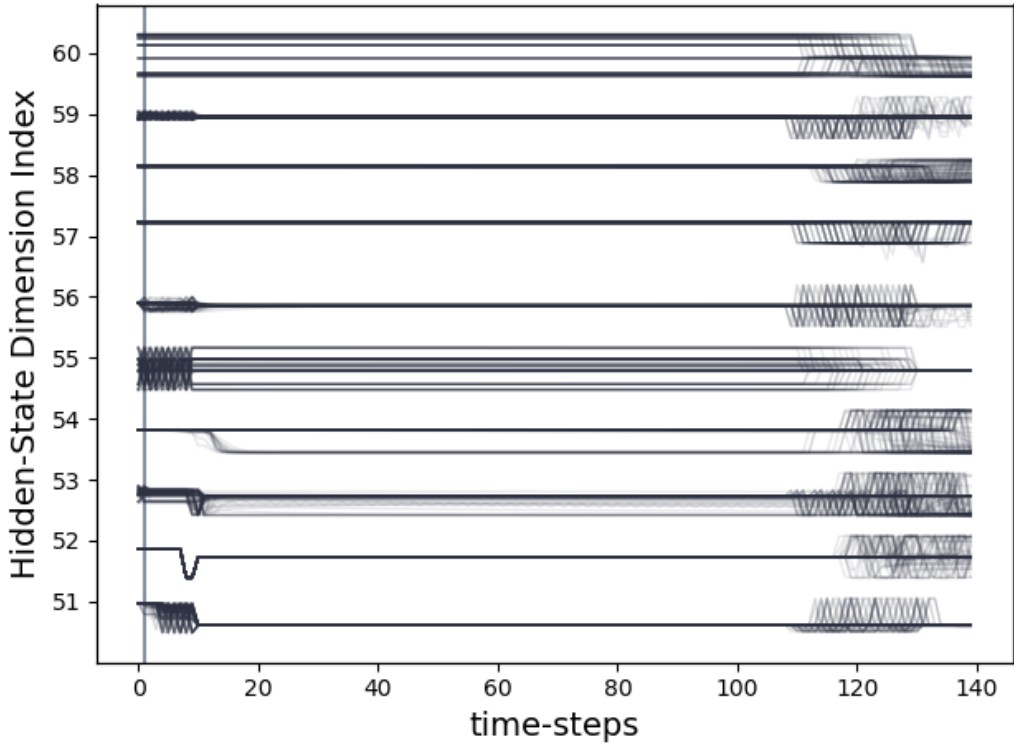

Figure 20: Superimposed trajectories one thousand trials of VDCM, from a perfectly trained GRU.

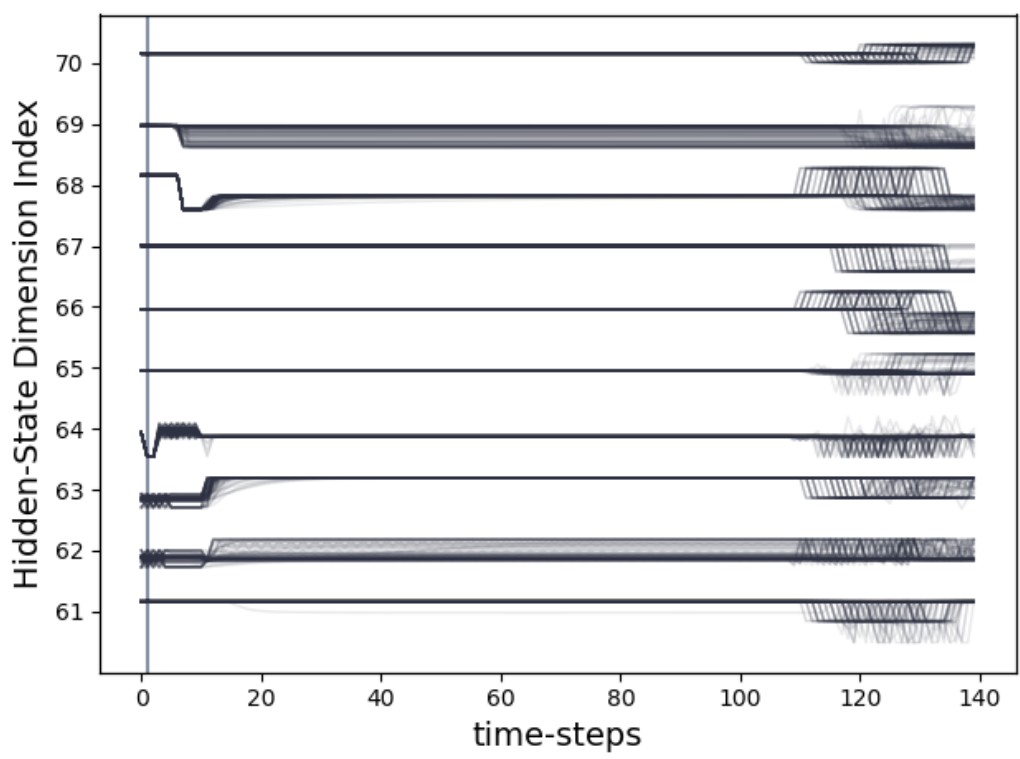

Figure 21: Superimposed trajectories one thousand trials of VDCM, from a perfectly trained GRU.

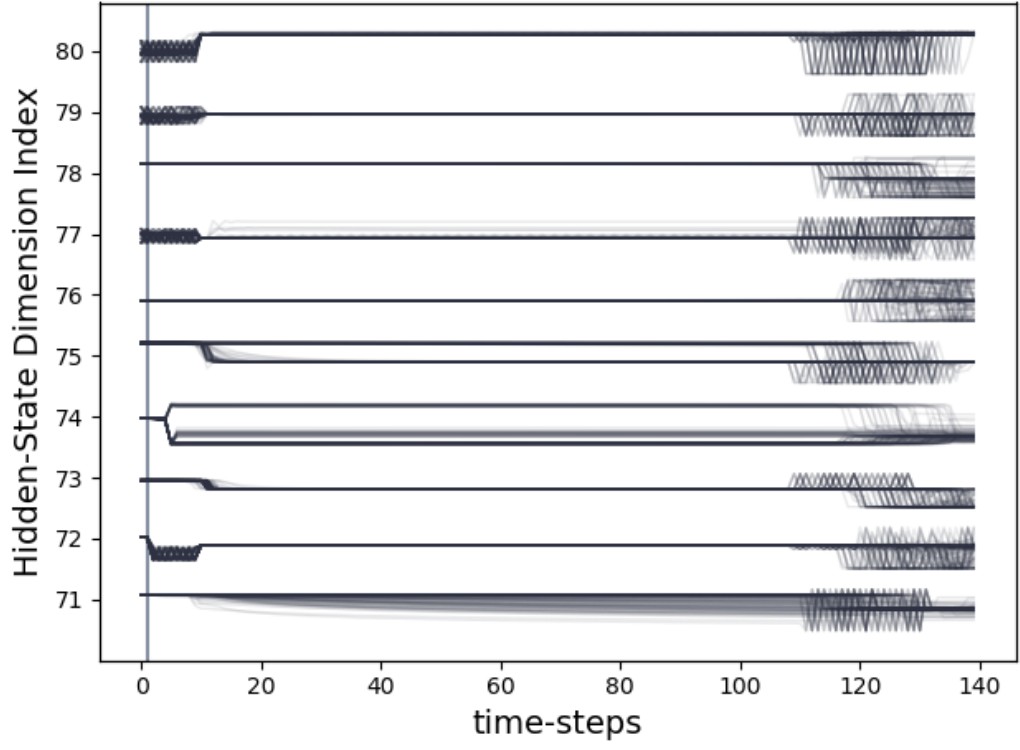

Figure 22: Superimposed trajectories one thousand trials of VDCM, from a perfectly trained GRU.

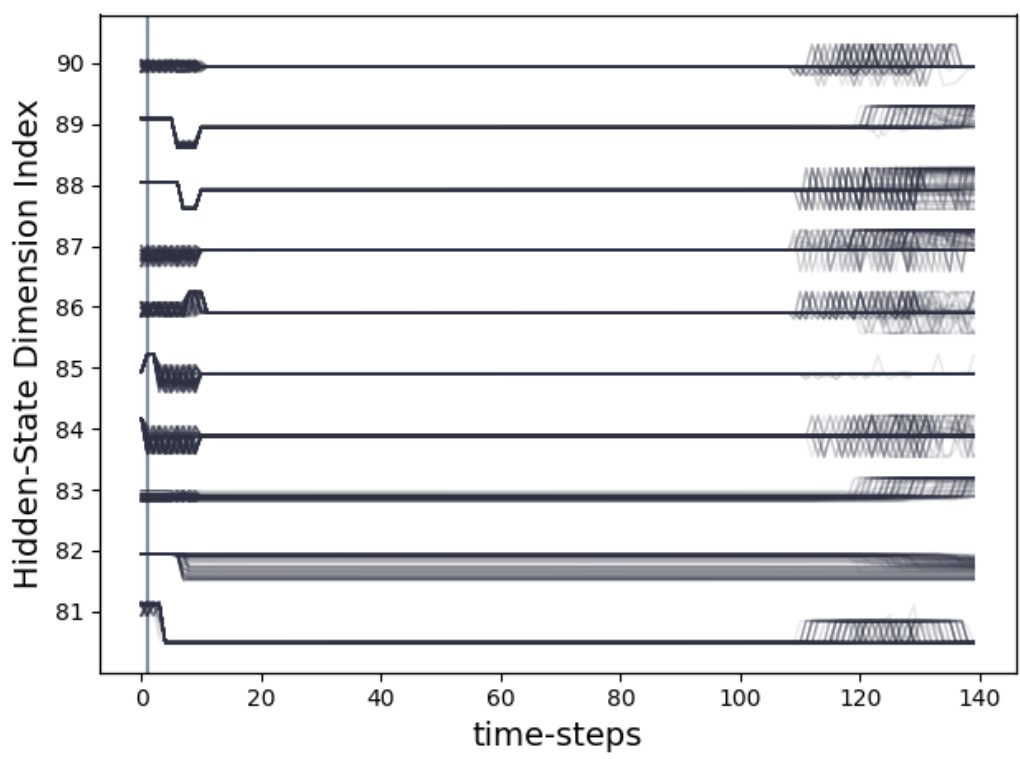

Figure 23: Superimposed trajectories one thousand trials of VDCM, from a perfectly trained GRU.

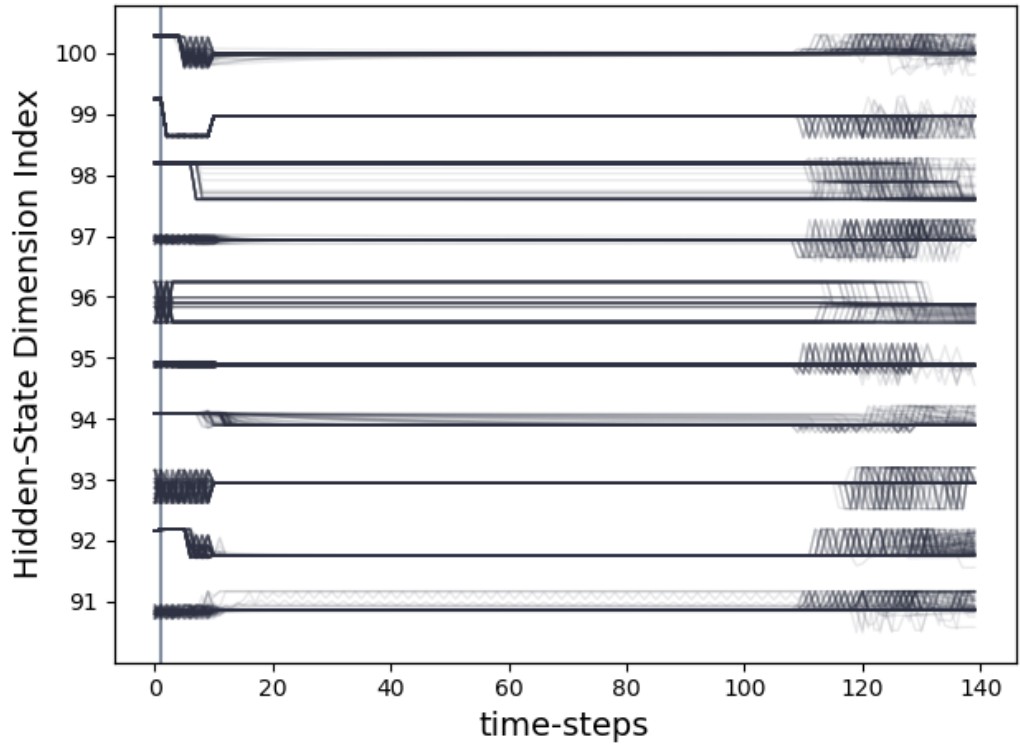

Figure 24: Superimposed trajectories one thousand trials of VDCM, from a perfectly trained GRU.

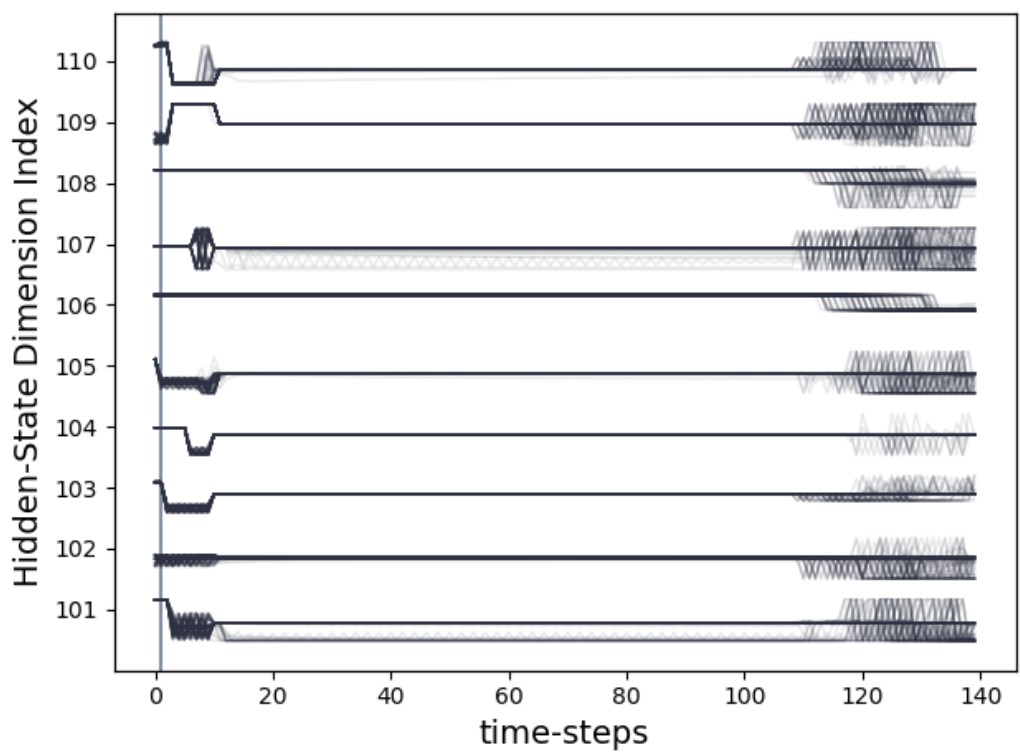

Figure 25: Superimposed trajectories one thousand trials of VDCM, from a perfectly trained GRU.

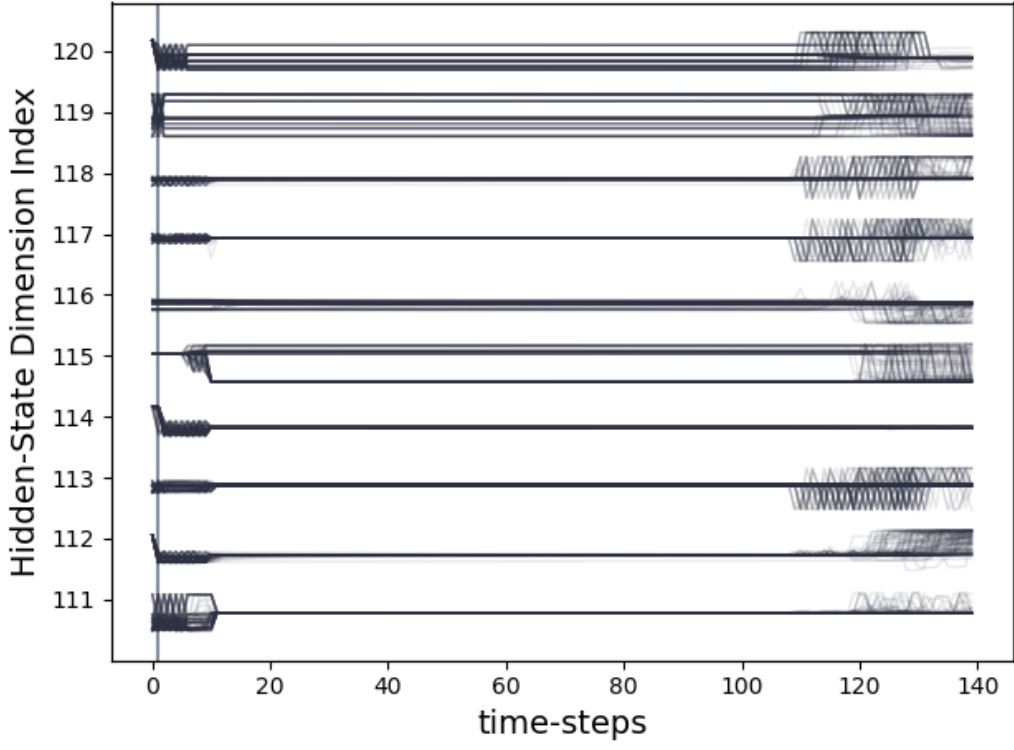

Figure 26: Superimposed trajectories one thousand trials of VDCM, from a perfectly trained GRU.

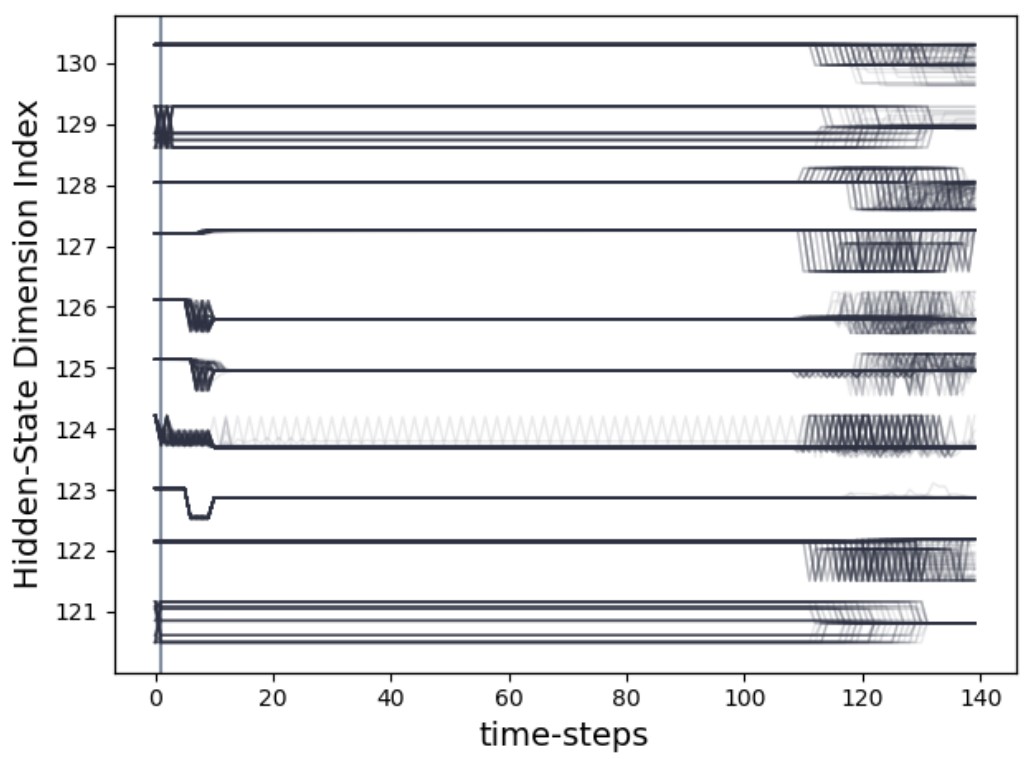

Figure 27: Superimposed trajectories one thousand trials of VDCM, from a perfectly trained GRU.

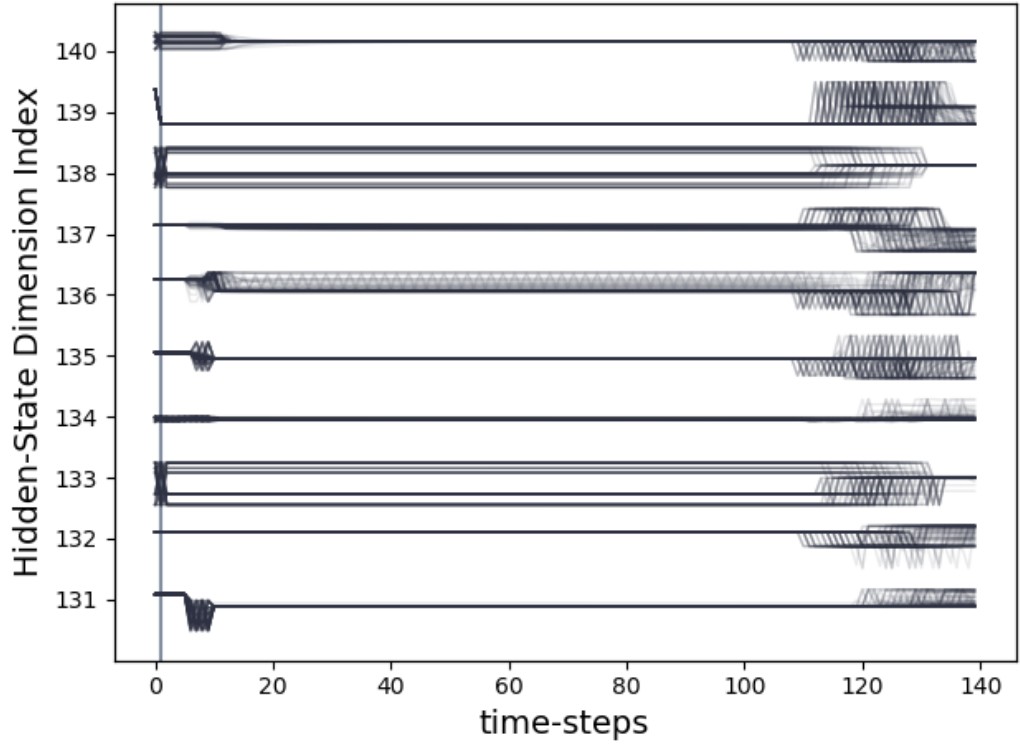

Figure 28: Superimposed trajectories one thousand trials of VDCM, from a perfectly trained GRU.

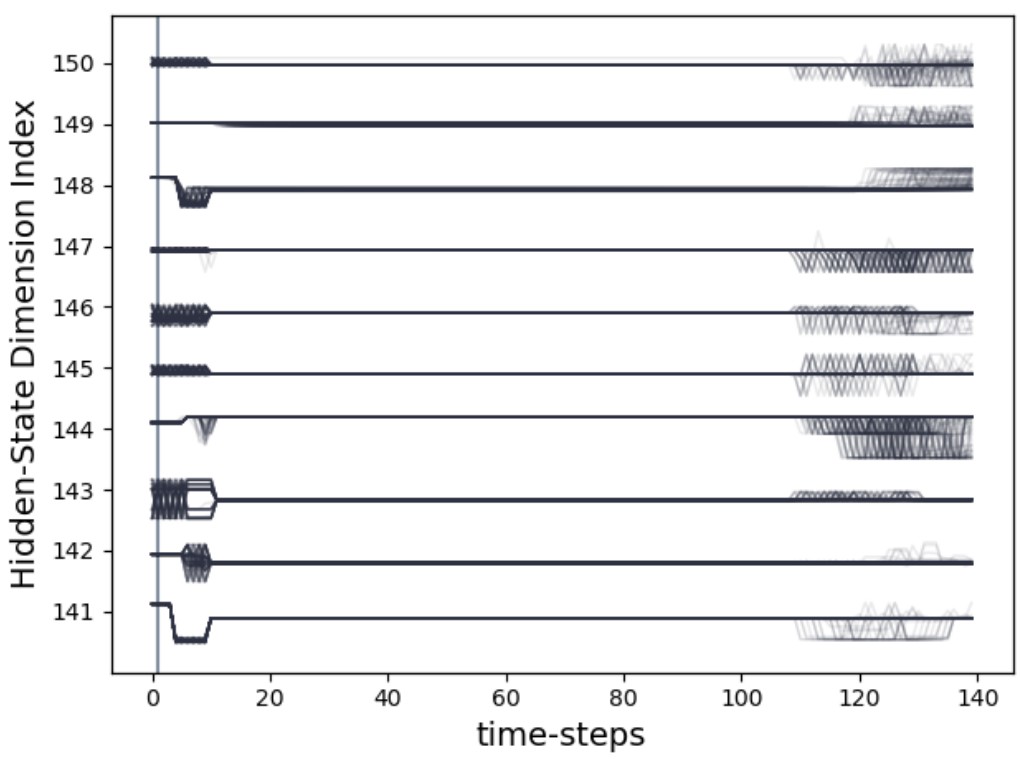

Figure 29: Superimposed trajectories one thousand trials of VDCM, from a perfectly trained GRU.

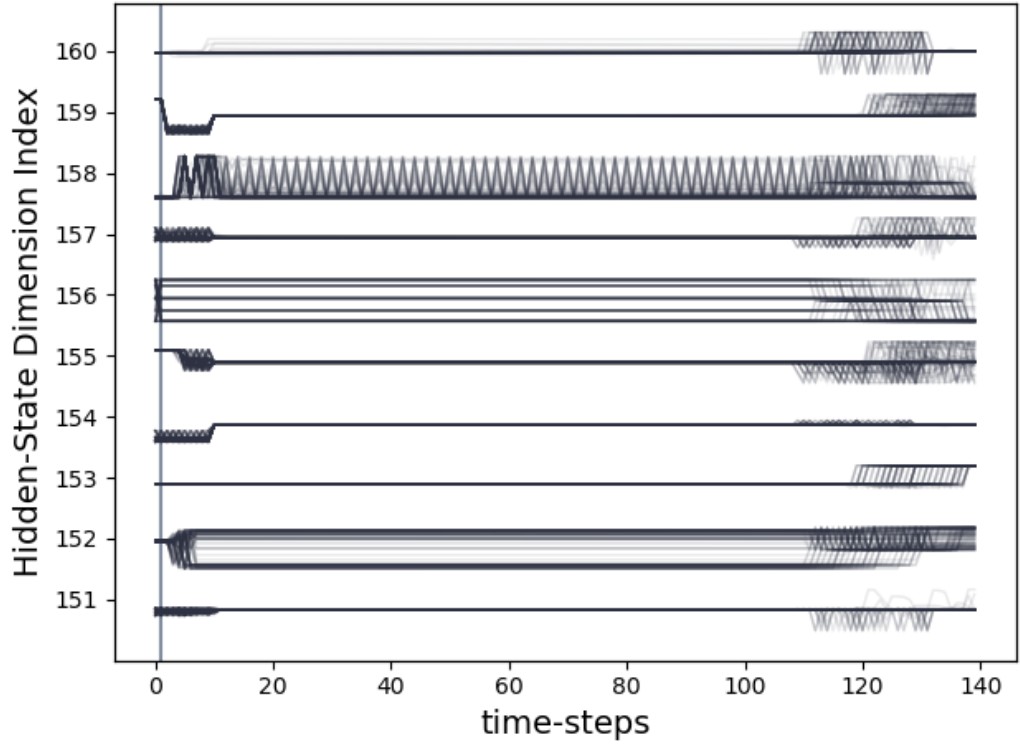

Figure 30: Superimposed trajectories one thousand trials of VDCM, from a perfectly trained GRU.

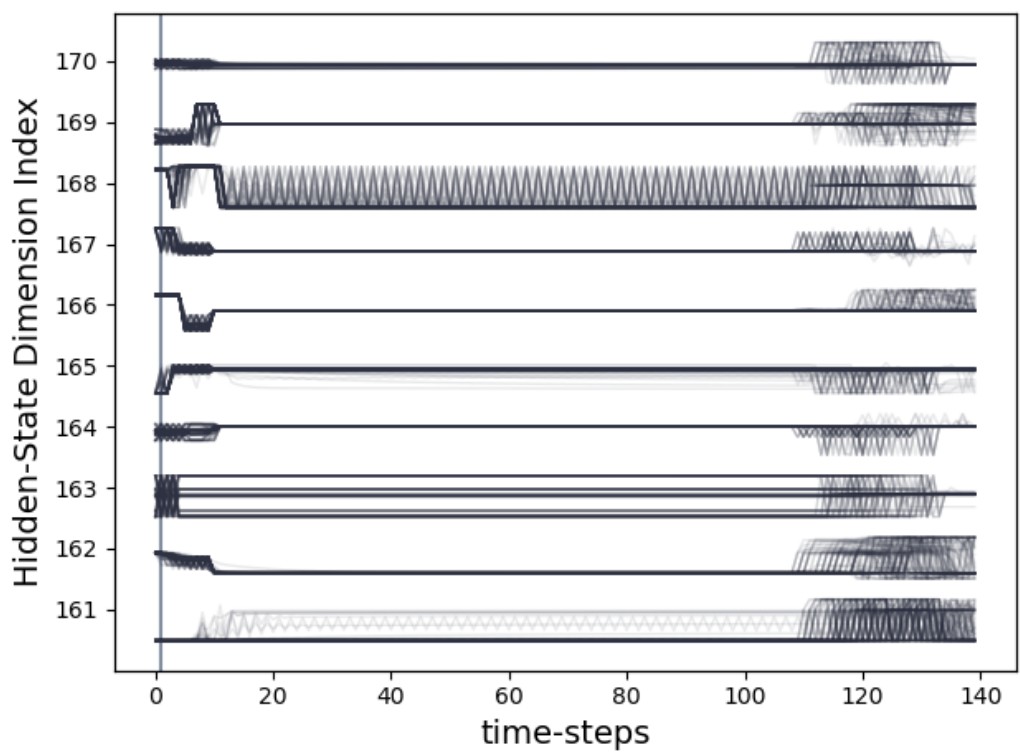

Figure 31: Superimposed trajectories one thousand trials of VDCM, from a perfectly trained GRU.

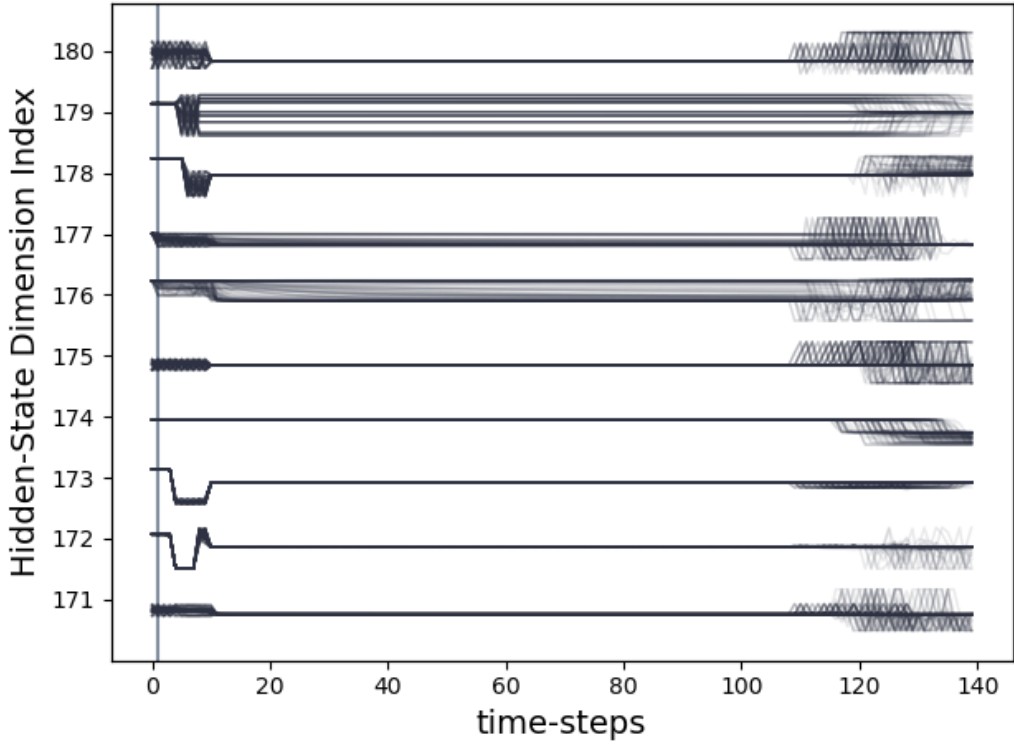

Figure 32: Superimposed trajectories one thousand trials of VDCM, from a perfectly trained GRU.

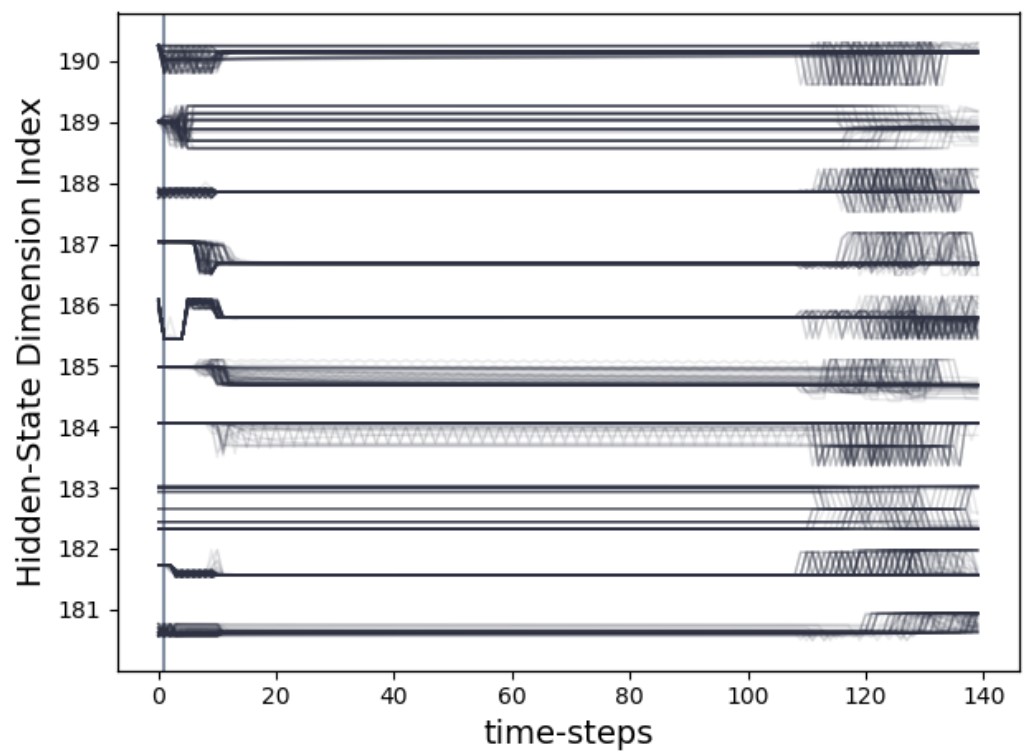

Figure 33: Superimposed trajectories one thousand trials of VDCM, from a perfectly trained GRU.

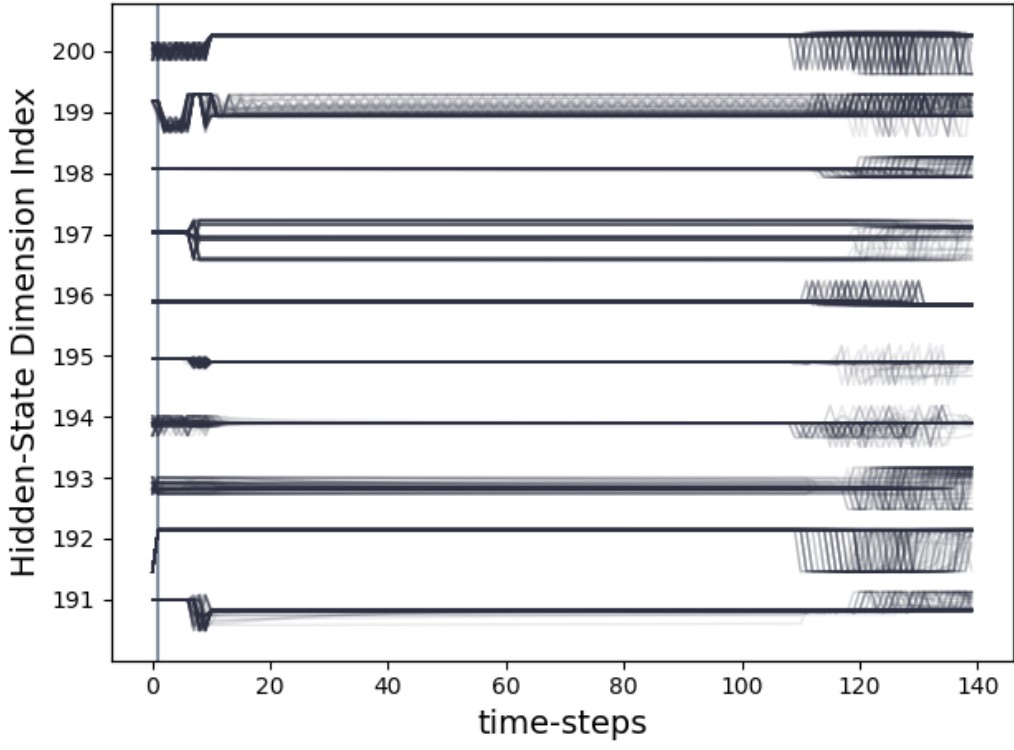

Figure 34: Superimposed trajectories one thousand trials of VDCM, from a perfectly trained GRU.

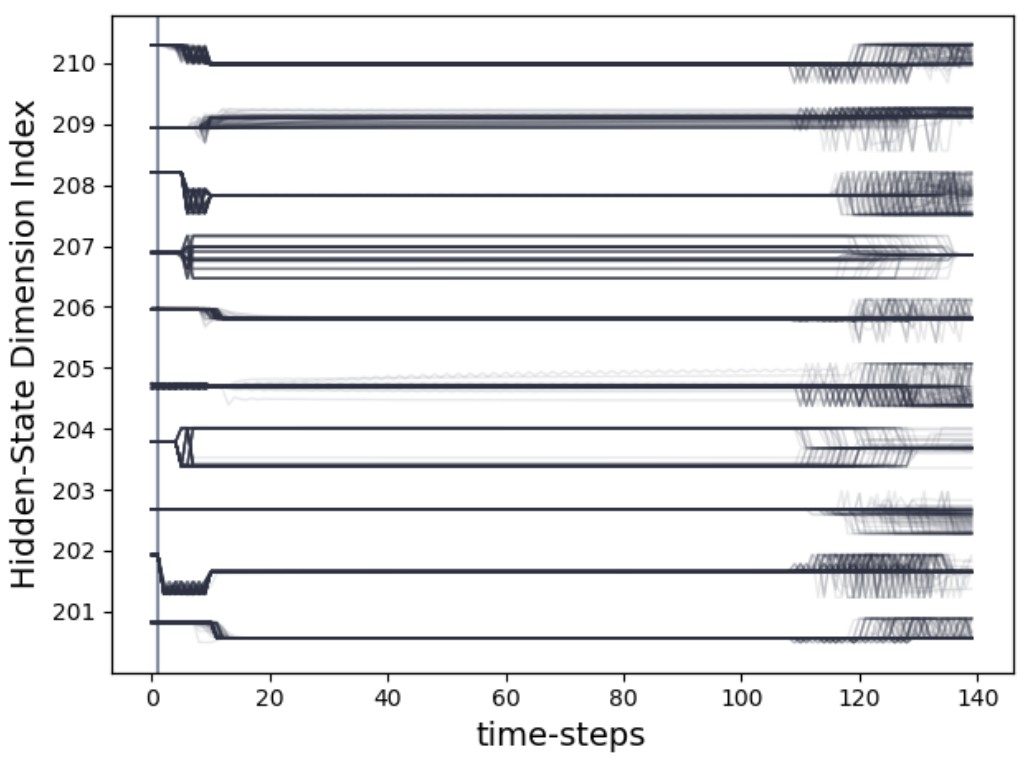

Figure 35: Superimposed trajectories one thousand trials of VDCM, from a perfectly trained GRU.

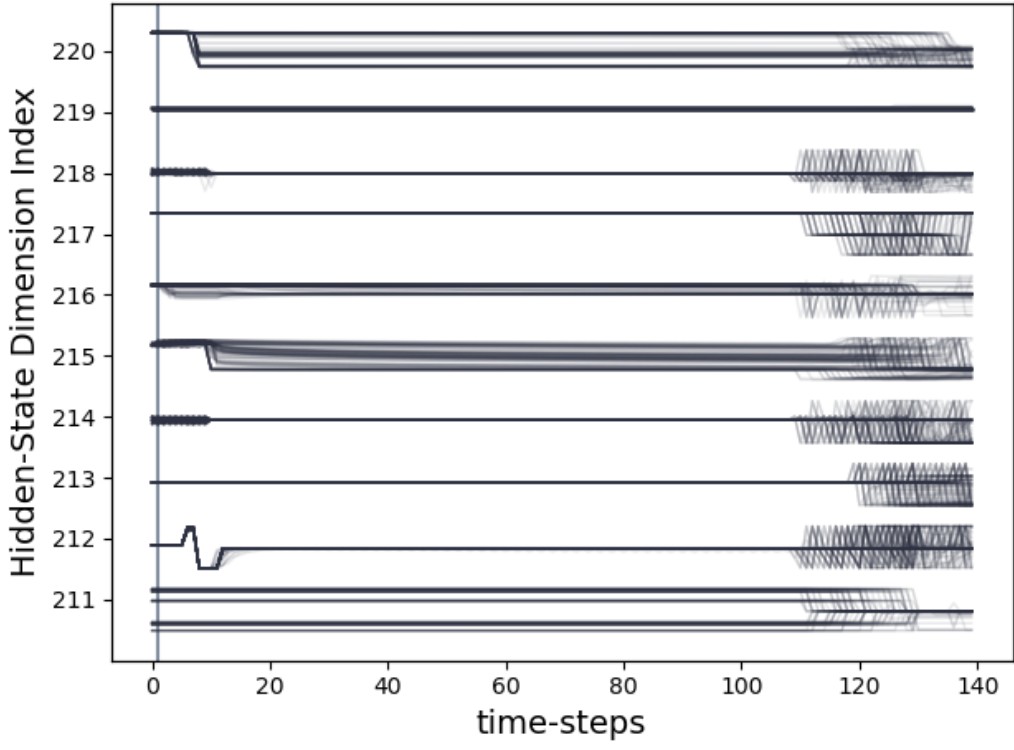

Figure 36: Superimposed trajectories one thousand trials of VDCM, from a perfectly trained GRU.

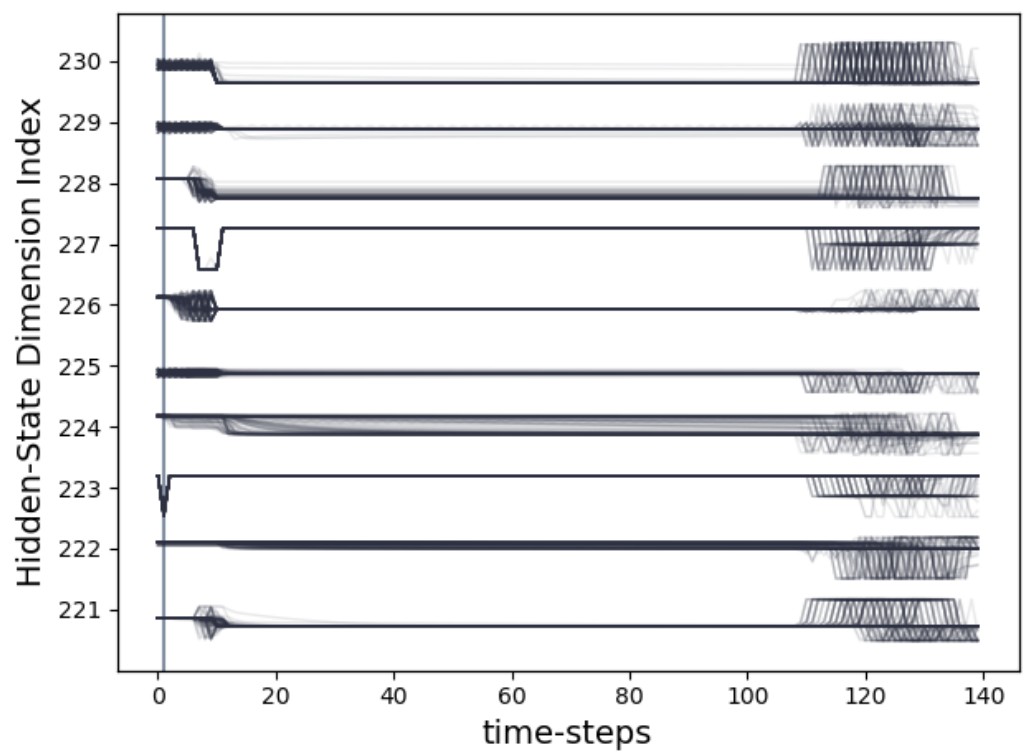

Figure 37: Superimposed trajectories one thousand trials of VDCM, from a perfectly trained GRU.

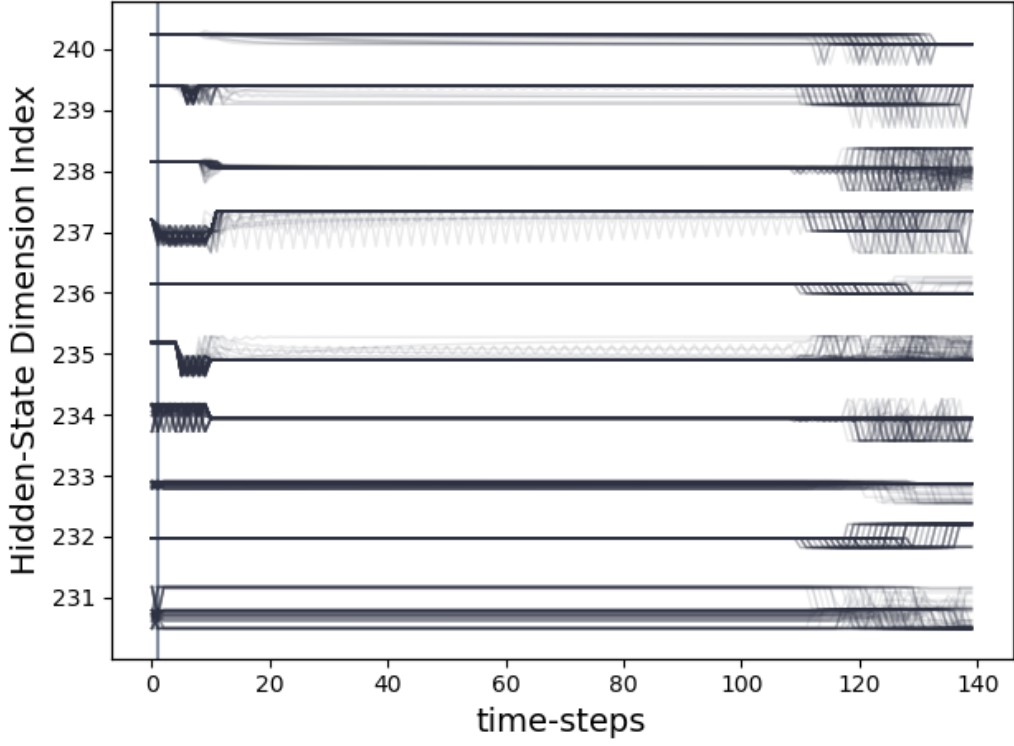

Figure 38: Superimposed trajectories one thousand trials of VDCM, from a perfectly trained GRU.

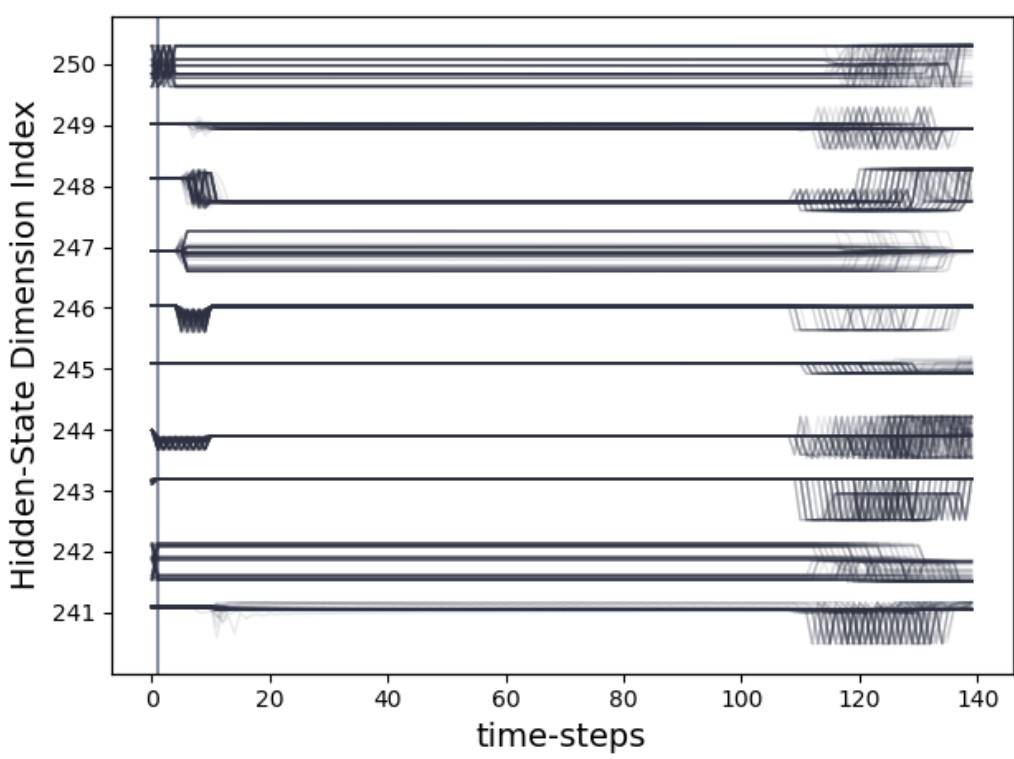

Figure 39: Superimposed trajectories one thousand trials of VDCM, from a perfectly trained GRU.

