# OpenReview forum: "Gating Mechanisms Underlying Sequence-to-Sequence Working Memory"
_ICLR.cc/2022/Conference — ICLR 2022 Submitted_

### Official Review · Reviewer_AM8u · 2021-10-29

**Correctness:** 4
**Technical Novelty And Significance:** 3
**Empirical Novelty And Significance:** 4
**Recommendation:** 8
**Confidence:** 3

**Main Review:**

#### Strengths:
- Thorough experimentation and exciting insights.
    - Through perturbation analysis, the paper finds that only 73 of 250 state elements encode memory.
        - PCA analysis solidifies this finding in Figure 4.
    - A reset measure $\kappa$ is defined in section 4, the new measure allows visualization of decoding resets in Figure 5.
- Using the discovered mechanisms the paper proposes a synthetic solution to the variable length memory problem.
- Source code with a few docstrings is available.
- The paper is a good fit for ICLR.


#### Weaknesses:
- Related Work: Previous work by Karpathy et al. studies the behaviour of various recurrent cell gates across recurrent architectures for natural language processing tasks. The study appears to be relevant for this article. Authors may want to be fair and consider a citation?
    - Karpathy, Andrej et al. "Workshop Track - ICLR 2016 Visualizing and Understanding Recurrent Networks." (2016).

- The description of the experiments was often hard to read. Speaking of 'neurons' instead of reset-gate-neurons,
  update-gate-neurons or hidden-state-neurons, for example, made it harder at first.
  Readers can infer the exact setting from context but should not have to do that.

#### Questions:
- Figure 1: Hidden state neurons mean we are looking at the output from the tanh in equation (3)?
- Figure 2: I personally found it tricky to understand what is going on here.
    I could follow the description of the argmax,
    but what is the difference between the "Projection to hyperplanes from hidden state" and the white points?
    If the white points are the entries of $\mathbf{y}_0, \dots , \mathbf{y}_n $$ aren't these points the projection?
    Why is the projection along the value axis?
- Figure 3, 4: We are looking at hidden neurons, aren't we?
- I think neuron is always shorthand for hidden-neuron?
  If that is true, the authors may want to consider explicitly stating this somewhere?

#### Minor remarks:
- Figure 4, caption: ...every eight column corresponds... ( switch plural columns to singular )

**Summary Of The Paper:**

The paper studies GRU cell dynamics on the variable-length memory problem in depth.
Building on top of the uncovered mechanisms a synthetic solution is proposed.

**Summary Of The Review:**

To the best of my knowledge, the paper appears experimentally sound and different from previous work.
I firmly believe that we need more work trying to uncover the inner workings of neural networks.
I am therefore recommending acceptance in the proceedings, assuming that the differences to the analysis from Karpathy et al. (citation in main review) will be discussed briefly.

---

> ### Author Response · Authors · 2021-11-18
> **Response to Reviewer**
>
> We thank the reviewer for their positive and insightful review of our manuscript. We’ll begin by responding to the listed weakness in the main review. After reading the suggested citation (Karpathy et al., 2016), we agree that this study is relevant to our work. We’ve included a short discussion of the differences between their manuscript and our own in the introduction of our resubmission. Regarding the writing being difficult to follow at times, this was a unanimous opinion across reviewers. We’ve taken the time to clean up and further elaborate all the passages we and or the reviewers felt would benefit from editing. Furthermore, we’ve ensured that the conventions in our language are easy to follow.
>
> Responses to further questions (the manuscript has been altered to address these questions and to improve the reader’s understanding):
>
> - Hidden-state neuron refers to the value h_t on the LHS of equation 3.
> - we can express each element of y_t as a linear combination of the values of each neuron (defined by the learned output weights), which can be interpreted as a hyperplane, where our position on this hyperplane is determined by h_t. The hyperplanes can be arranged such that regions in the hidden-state space exists, where each possible readout takes on a higher value than the others. The white points indicate the class decision boundaries, and the blue points indicate each element of y_t from an example h_t.
> - Yes, in figures 3 and 4 we are looking at hidden neurons
> - We have defined this shorthand naming convention explicitly early in the manuscript
>
> - In regards to the minor remark, we have corrected the typo.

---

> > ### Comment · Reviewer_AM8u · 2021-11-24
> > **Comment on revised version and other reviews.**
> >
> > I have read the other reviews and the revised version of the paper. Regarding the presentation, I agree with reviewers MToe, afLb and TN5R. The clarity of the writing needed improvement.
> > The issues I have raised have been fixed in the revised version. The authors have resolved the ambiguity regarding the hidden neurons in the text. A discussion of the differences with respect to Karpathy et al., 2016 is now part of the related work.
> > I think this work is different enough to stand on its own.  I am satisfied with the changes and continue to recommend inclusion in the proceedings.

---

### Official Review · Reviewer_MToe · 2021-11-02

**Correctness:** 3
**Technical Novelty And Significance:** 3
**Empirical Novelty And Significance:** 1
**Recommendation:** 6
**Confidence:** 3

**Main Review:**

The paper shed light on the encoding and decoding process of GRU, explaining its capability and behaviours. Overall, the authors' hypotheses are well-validated through empirical evidence and informative visualization of the synthetic task. Among the findings,  the synthetic solution is the most interesting and gives novel insights into GRU's operation. On the other hand, the writing is hard to follow. Here, the paper limits to demonstrating the phenomenon and leave the action for future work, which is somewhat incomplete.

Questions and comments:
- The term "slow manifold" is mentioned frequently without any explanation. From the writing, it seems to simply refer to the fact that during the delay phase, some neurons change slowly. To make the paper self-contained, a brief definition would be appreciated.
- Algorithm 1 is critical for readers to understand Section 3. Please describe it in more detail in the main manuscript. Or maybe add a simplified version of Algorithm 1.
- Section 3 seems to suggest that GRU "memorizes" all pairs of step-element during the encoding phase. If so, it cannot generalize to test data with a slightly different S. Did the authors test this case?
- Fig. 3 is hard to read. Top's y-axis: what does "accuracy" mean? Is there any meaning in the point's colour?
- Fig. 4-left, what is the meaning of the big horizontal line at neuron 60?
- Fig. 5-left, the label K is confused with the number of symbols
- Fig.5-right, any explanation for the exception at t=3? How does the GRU perform at this step? If it still decodes correctly at t=3, what makes it possible?
- Sec. 5, did you verify your synthetic solution by comparing it with the parameters of the trained GRU. Is this possible that, after training, GRU can converge to your synthetic solution?

**Summary Of The Paper:**

The paper analyzes GRU's underlying mechanisms that store and retrieve information in the delay copy task of a sequence of K symbols. It proposes a perturbation-based method to determine which neurons are responsible for encoding a step-element pair. The paper then shows that at each step of the decoding phase, certain neurons are reset by GRU, generally mapping to those tuned to hold information of step-element pair.  Finally, the paper provides a synthetic solution to the delay copy task for the case K=2.

**Summary Of The Review:**

An interesting paper, yet the writing needs improvement. It is important to understand the underlying mechanism of models like GRU. However, I am unsure if the presented result is useful enough for designing a better mechanism for RNN/GRUs.

---

> ### Author Response · Authors · 2021-11-18
> **Response to Reviewer**
>
> We thank the reviewer for a positive and informative review. In regards to the difficulty in understanding parts of our writing, we apologize and have taken the steps required to improve the readability of our manuscript.
>
> Responses to posed questions and suggestions (the manuscript has been altered to address these points and improve the reader’s understanding):
>
> - An explicit definition of the slow manifold has been added to the manuscript. The reviewer’s assumptions of the dynamical feature are correct. A slow manifold is similar to an attractor (i.e. a fixed point, an attracting line, an attracting ring, etc.), where if the state of the system h_t lies on the attractor it will not change unless perturbed. However, the manifold is not entirely made up of fixed points, rather the speed of the phase flow in these regions is arbitrarily slow in a subset of directions. The GRU has two primary mechanisms to achieve such behavior, which are (1) pseudo-line attractors, and (2) via influence from the update-gates associated with each hidden-state neuron in the network (Jordan et al., 2021).
> - Further detail on Algorithm 1 will be added to the main text.
> - We have tested if the network can generalize to different values of S. It appears that the internal clock learned in the network is specific to S=10. If we vary S, not a single trial’s readout is entirely correct. If S < 10, the first S elements are properly encoded/decoded. If S > 10, the first 10 elements are properly encoded/decoded. These details have been added to the main text of the manuscript.
> - We better explain figure 3 to be more understandable, along with our explanation of Algorithm 1. “Accuracy” refers to how successful each perturbation to h_t made at t=11 (i.e the beginning of the delay period) was across the test data set. In other words, if the perturbation made was intended to change the first time-step of decoding to a “3,” for example, and leave all other time-steps unaltered, and does that consistently with no mistakes for every trial in the test set, the accuracy is 1, implying 100% of the test set trials were successfully altered by the perturbation at t=11. The color indicates which element was intended to be read out at the designated decoding time-step.
> - The hidden-state neurons (rows) contained in the green shaded region in Figure 4 (left) are used to create the PCA plot, Figure 4 (right). This point is made more apparent in the figure caption.
> - The font in Figure 5 (left) for the Greek letter kappa, has been altered to distinguish it better from the letter “K.”
> - We assumed that the mechanisms for decoding each individual time-step were not unique, and were actually quite surprised when we found that the trained network decodes all but those at t=3 essentially the same way. However, such a decoding step is very possibly a more complex variant of the same behavior at all the other time-steps. If we look closer at Figure 5 (right), it appears as though the Information from t=3 is reset after the fourth time-step of decoding, for some values of K, suggesting possible intersection between the regions of the slow manifold , and or that the geometry of theses regions are more complex, requiring the neurons tuned to additional time-steps to aid in enacting the computation.
> - The parameters of the synthetic solution, while functionally inspired by the trained network, do not resemble the parameters of  that network, that fall primarily into a Gaussian-like distribution with a high kurtosis. The parameter matrices of the synthetic solution are extremely sparse. To the best of our knowledge, we’ve never seen such an interpretable solution manifest for a task of this complexity through a gradient based training regime. While maybe not impossible, we don’t think one should expect such a result unless the training incorporated highly specific constraints derived from the synthetic solution demonstrated  in this paper.

---

### Official Review · Reviewer_afLb · 2021-11-02

**Correctness:** 4
**Technical Novelty And Significance:** 1
**Empirical Novelty And Significance:** 1
**Recommendation:** 3
**Confidence:** 4

**Main Review:**

In terms of strengths, I think the authors definitely show that for this specific task and architecture, the GRU is learning to use working memory in a more or less explainable way.

In terms of weaknesses, I feel that this paper lacks novelty. Their work is not technically novel because they are using a GRU and using a super well studied task. I don't think I took away any sort of new information from their analysis either. Hence I am not sure what the contribution of this work is to the community. I would also like to comment on how busy the author' figures and experiment descriptions are and how hard they are to follow.

If the authors wanted to improve the novelty of their work, they can maybe focus on state of the art architectures and more complex tasks.

**Summary Of The Paper:**

This paper attempts to understand how a RNN goes about solving the Variable Copy Delay Memory task in fine detail. Their model has been trained perfectly on this task and so they are able to focus on how it goes about changing its cell values in the case. The authors present different metrics to track resetting and memorization behavior by the GRU's neurons.

**Summary Of The Review:**

This work lacks novelty, is hard to follow, and shouldn't be included in ICLR.

---

> ### Author Response · Authors · 2021-11-18
> **Response to Reviewer**
>
> We respectfully disagree with the reviewer’s claims that (1) the GRU architecture is not worth studying, and (2) that “variable delay copy memory” (VDCM)  is a well studied task. In the case of the architecture, the GRU was made as a simplification of the LSTM. However, we were unable to train an LSTM network to accurately perform VDCM (Figure 12 -- Supplementary material). Additionally, we will include training curves for the newer coRNN and Lipschitz RNN architectures to the updated manuscript. Furthermore, no finite memory system can solve a task of this type, including but not limited to transformer networks and reservoir architectures. In the case of VDCM, the GRU appears to have a distinct advantage in learning the necessary behavior required to enact the desired computations, thereby accurately performing the task. We have a separate manuscript, currently in preparation, that explains this phenomenon.
>
> Furthermore, to the best of our knowledge, no one has ever studied VDCM at this low a level of analysis. If this is not the case, we kindly ask the reviewer to provide us with the appropriate citation(s). It is clear from the reviewer’s take on our paper, that there were miscommunications regarding the novelty of our work. We’ve made the necessary adjustments, to both better elaborate on these points, and to improve the overall readability of our manuscript.
>
> Please consider these points and re-evaluate the manuscript for the impact and significance of our findings. ICLR is dedicated to advancing the theory of deep learning, be it standard or non-standard. While many are focused on the state-of-the-art, we as a community do not fully understand what came before. It is difficult to know where innovation will stem from, and we believe filling in these gaps of understanding will better equip researchers, you and us alike, to tackle the ever growing list of problems in the coming years.

---

### Official Review · Reviewer_TN5R · 2021-11-02

**Correctness:** 3
**Technical Novelty And Significance:** 2
**Empirical Novelty And Significance:** 3
**Recommendation:** 6
**Confidence:** 4

**Main Review:**

Studying the mechanism by which an RNN model stores and retrieves memory is an important question that can give insights into how these networks work.

This paper's approach of trying to reverse engineer the mechanism of a trained GRU to understand this is interesting. Moreover the finding that slow manifolds are used for the purpose of storing memory of a sequence is interesting.

But the paper does not consider or test alternate hypotheses other than slow manifolds. For example, no attempt is made to empirically show that the GRU is not using pseudo-line attractors. This is a major flaw of the paper.

It is also not clear why the authors specifically consider the GRU architecture, especially since it seems like using an LSTM would make it easier to reason about when information enters and exits the cell.

The writing of the paper is also very confused and hard to parse in many places. The sequence of arguments made for slow-manifolds is sometimes hard to follow.

For example:
- Only 3 units are shown from the trained GRU, with text indicating that this is representative of all other neurons. But later the authors say only 73 neurons were analysed. Might be a good idea to show the plots for all neurons, possibly in the supplement.
- It might help to explain formally what a slow manifold is.
- The perturbation based experiment is hard to understand, as is the PCA analysis.
- Minor: Fig. 3: what are the colors?

**Summary Of The Paper:**


The authors study the mechanisms that a GRU network uses to store and retrieve memory in a variable delay copy task. the authors show that the GRU uses slow manifolds to do this, and construct a network for a smaller variant of the task by hand.

**Summary Of The Review:**

Overall, while the goals of the paper are well-founded, and their analysis seems potentially interesting, a combination of lack of clarity and some gaps in their overall argument weakens the paper significantly.

---

> ### Author Response · Authors · 2021-11-18
> **Response to Reviewer**
>
> We thank the reviewer for an informative and honest review of our manuscript. We are happy to see that the reviewer agrees that the questions we set out to answer are interesting along with our methods. Given the criticisms in the main review, it is apparent that we miscommunicated several points in the paper. We will be responding to these points here, and have updated the manuscript accordingly to better reflect what we mean and improve readability.
>
> The first criticism is that we neglect alternative hypotheses for how information is encoded in the network, other than slow manifolds. The reviewer comments on us ignoring the use of pseudo-line attractors in the trained network as an alternative hypothesis. As mentioned in section 3 of the paper, the GRU has two means to achieve slow manifolds: (1) via the update-gate, and (2) via pseudo-line attractors, as discussed in (Jordan et. al., 2021). To incorporate pseudo-line attractors into our analysis would still be a discussion on slow-manifolds, just achieved by different means. We, however, may not have been clear as to why we neglected pseudo-line attractors all together. Such slow flow is the result of the nullclines of the underlying continuous-time dynamical system, by which the network can be interpreted as a numerical approximation of, existing sufficiently close together in the hidden state-space. However, the nullclines of this system cannot be oriented such that they form a pseudo-line attractor in any canonical direction. As such, if they were used to encode information, the use of the reset-gate as part of the decoding mechanisms would be rendered insufficient, as the GRU can only reset information in canonical directions, by design. Given that our decoding analysis worked well, it did not seem necessary to dive further into alternative decoding schemes that would satisfy the use of pseudo-line attractors.
>
> As for alternative hypotheses outside of slow manifolds, the behavior of the hidden-state neurons during the delay period (Fig. 1) rule out any dynamic encoding schemes (periodic oscillations, quasi-periodic oscillations on various manifolds, finely tuned fractional dimensional attractors, etc.). This leaves us with either continuous-attractors, which are hypothesized to not exist in any finite dimensional GRU network (Jordan et al., 2021), fixed-points, which would be highly non-trivial to orient given the sequential resetting of information from the reset-gate during decoding, or slow-manifolds.
>
> The second criticism is that it is unclear why the GRU architecture was chosen, and why an LSTM network was not considered. Simply put, we wanted to dissect a network that performed variable delay copy memory (VDCM) perfectly -- zero mistakes on the test set. We were unable to train an LSTM network to do this (Figure 12 -- in supplementary material). Of all the network architectures, initializations, and hyperparameter combinations we tried, only the GRU network using an original initialization scheme of ours was able to achieve this level of accuracy. We have a manuscript in preparation centered around this initialization technique, which brings to light why such a scheme works.
>
> The third criticism is in regards to the clarity of our writing. We apologize for the difficulty in making sense of our work. We have gone back through our manuscript, and have fixed the example points brought across reviewers. The following are some of the specific changes made, per suggestion of the reviewer:
>
> - All of the hidden-state neurons have been plotted in the appendix, as extension to figure 1. To clarify, the three neurons shown in figure 1 are qualitatively representative of all basic function of those in the network, as neuron #84 does not demonstrate slow-manifold dynamics and is a good candidate for a neuron not used in the slow-manifold (i.e. not one of the 73 mentioned neurons). We’ve better explained this point in the main text of the manuscript.
> - An explicit and detailed definition of a slow-manifold has been added to the manuscript.
> - We’ve reworded and better explained the perturbation analysis and PCA analysis, in order to make our work more accessible and easier to understand.
> - The colors in figure 3 represent the chosen K for each time-step. We’ve made mention of this point, to be more clear.

---

> > ### Comment · Reviewer_TN5R · 2021-11-29
> > **Concerns addressed**
> >
> > I thank the authors for their detailed response and revision. Seeing the plot of all the neurons really drives home the message that a slow-manifold mechanism is used by most neurons. And with the improved explanations, I am able to follow the authors arguments, and agree that this is insightful work. Being able to analytically construct solutions using this understanding is also very interesting.
> >
> > Since, most of my concerns are addressed now, and so I will increase my score to an accept.
> >
> > There are a few relatively small suggestions that I would request the authors to incorporate into their paper if possible:
> > - (Abstract) Not true that reservoir networks can't achieve memory retention and manipulation, since the reservoir units can easily be units that have longer memory capabilities. See e.g. [1]
> > - Section 3: Explain why "Only 73 of the 250 neurons in the network may be used to encode memory." Do you mean, they "are" used to encode memory, or "can be" used to encode memory? Why?
> > - "We were unable to train other network architectures on VDCM, including LSTM." -- should add "to achieve perfect accuracy"
> > - The figure in Appendix E could optionally be made smaller. Fig. 17 and 18 seem to be accidentally identical.
> >
> > [1] Salaj, Darjan, Anand Subramoney, Ceca Kraisnikovic, Guillaume Bellec, Robert Legenstein, and Wolfgang Maass. ‘Spike Frequency Adaptation Supports Network Computations on Temporally Dispersed Information’. ELife 10 (July 2021): e65459. https://doi.org/10.7554/eLife.65459.

---

### Decision · Program_Chairs · 2022-01-20

**Decision:**

Reject

**Comment:**

This paper studies how recurrent neural networks, and more specifically GRUs, store and access information. The authors analyze the solution obtained by gradient descent to the variable delay copy memory task for discrete sequences. They use concepts from dynamical systems, such as slow-manifold, to understand the behavior of the learned model. Finally, based on this analysis, the authors propose a synthetic solution to a simplified version of the delay copy memory task.

Overall, while the scores for the paper are rather positive, I still have concerns about the paper, based on the reviews and discussion. I do not believe that these concerned were well addressed by the authors in their rebuttal. First, I tend to agree that the paper is somewhat lacking novelty and insightful findings (reviewers TN5R, afLb, MToe). For example, I think that tools from dynamical systems are mostly useful to analyze RNNs when the input is constant (Jordan et al., 2019). In the case of the copy task, this corresponds to the "delay" period, where in practice the hidden state is almost constant. This behavior is easily explained by the value of the update gate, close to 1. I thus agree that other hypotheses than slow manifold should be discussed to explain how GRUs store and access information, and that the benefits of using dynamical systems is not obvious. Moreover, I believe that previous solutions to the copy task (eg, from Henaff et al.) could be extended to the variable setting by adding a gating mechanism to these solutions. In particular, Henaff et al. claimed that LSTM could solve this task empirically, while the authors claim otherwise.

Second, after reading the revised version a couple of times, I still find the paper hard to follow (MToe, afLb, TN5R). For example, I think that the concept of slow manifold is not introduced properly, and in particular, how it applies to the learned solution is not clear. More generally, I found the sections regarding how information is stored and accessed a bit confusing. Finally, I think that the studied task is simple, and probably does not provide strong insight about the working of recurrent networks. Specifically, LSTMs tend to perform similarly or slightly better than GRUs on many tasks, while the authors claim that this architecture cannot solve the studied task.